# NOLC1 suppresses immunochemotherapy by inhibiting p53-mediated ferroptosis in gastric cancer

Shengsheng Zhao[1,2,3†], Ji Lin[1†], Bingzi Zhu[2†], Yin Jin[1], Qiantong Dong[1], Xiaojiao Ruan[1], Dan Jin[1], Yongdong Yi[2], Binglong Bai[2], Hongzheng Li[2], Danna Liang[2], Jianhua Lu[2], Letian Meng[2], Xiang Wang[1], Yuekai Cui[2], Yuyang Gu[2], Xian Shen[1,2], Xufeng Lu[2]*, Shangrui Rao[2]*, Weijian Sun[1,2,3]*

[1]Department of Gastrointestinal Surgery, The First Affiliated Hospital of Wenzhou Medical University Wenzhou, Wenzhou, China; [2]Department of Gastrointestinal Surgery, Zhejiang International Scientific and Technological Cooperation Base of Translational Cancer Research, The Second Affiliated Hospital and Yuying Children's Hospital of Wenzhou Medical University, Wenzhou, China; [3]Department of Colorectal and Anal Surgery, The First Affiliated Hospital of Wenzhou Medical University, Wenzhou, China

*For correspondence:
luxufeng@wmu.edu.cn (XL);
rsr120@126.com (SR);
fame198288@126.com (WS)

†These authors contributed equally to this work

## eLife Assessment

This **fundamental** study identified a novel role of NOLC1 in regulating p53 nuclear transcriptional activity and p53-mediated ferroptosis in gastric cancer. After major revisions, the evidence supporting the conclusions is **solid**. However, some new experiments are needed to draw more robust conclusions regarding the ferroptosis-associated studies.

**Abstract** Human gastric cancer (GC) is one of the most malignant cancers, and cisplatin (Cis)-based chemotherapy remains the main clinical treatment for GC. However, Cis resistance often occurs, largely limiting its therapeutic efficacy in tumors. Therefore, a better understanding of the drug resistance mechanism could reveal new approaches for improving GC treatment efficacy. Here, we define the integrative role of nucleolar and coiled-body phosphoprotein 1 (NOLC1), a molecular chaperone that is significantly upregulated in GC tissues and Cis-resistant GC cells. Knocking down NOLC1 increased GC sensitivity to Cis by regulating ferroptosis. Mechanistically, NOLC1 binds to the p53 DNA-binding domain (DBD), decreasing p53 nuclear accumulation stimulated by Cis and suppressing p53 transcriptional functions. Then, the p53-mediated ferroptosis is suppressed. Furthermore, the silence of NOLC1 promoted ferroptosis-induced immunogenic cell death (ICD) and reprogrammed the immunosuppressive tumor microenvironment, thereby increasing sensitivity to anti-programmed cell death-1 (PD-1) therapy plus Cis. The combination of anti-PD-1 plus Cis effectively inhibited GC growth without significant side effects. In summary, our findings reveal that targeting NOLC1 may be a novel therapeutic strategy for GC and may increase the efficacy of chemotherapy combined with immune checkpoint inhibitor (ICI) therapy.

## Introduction

Gastric cancer (GC) is the fifth most diagnosed cancer worldwide and the fifth most common cause of cancer-related death (*Siegel et al., 2024*). Since GC is usually diagnosed at an advanced stage, it

often has a poor prognosis (*Lordick et al., 2022*; *Smyth et al., 2020*). Cisplatin (Cis)-based chemotherapy remains the primary treatment for advanced GC (*Guan et al., 2023*). Chemotherapy not only has a direct cytotoxic effect but also contributes to the immunogenic cell death (ICD) and improves the efficacy of immunotherapy. Thus, immunotherapy, represented by anti-programmed cell death-1 (PD-1) therapy, plus chemotherapy, has become the standard for first-line treatment (*Lordick et al., 2022*; *Li et al., 2021*). However, the efficacy of chemotherapy is still restricted by chemoresistance, which is caused by gene mutations, gene inactivation, epigenetic modifications, signaling pathway changes, or cell metabolism disorders (*Song et al., 2023*); thus, the efficacy of immune chemotherapy is still limited in some patients (*Joshi and Badgwell, 2021*). Therefore, exploring novel targets of chemoresistance in GC to increase combination effectiveness is highly important for improving tumor prognosis and patient survival.

Ferroptosis, an iron-dependent form of novel regulated cell death (RCD), is regulated by various pathways, such as mitochondrial activity, oxidative stress, iron handling, and metabolism (*Jiang et al., 2021*; *Chen et al., 2021a*). Recently, studies have shown that activation of ferroptosis in GC is a promising strategy for overcoming GC resistance. For example, increased ATF2 expression can alleviate Cis resistance by regulating ferroptosis in GC (*Xu et al., 2023*). Additionally, GC cells can overexpress many negative ferroptosis regulators, such as *MIR522* and ACTL6A, to promote chemoresistance (*Yang et al., 2023*; *Zhang et al., 2020*). Moreover, ferroptotic cells can activate the immune response and reprogram the immunosuppressive tumor microenvironment (TME) via releasing various damage-associated molecular patterns (DAMPs), subsequently increasing the efficacy of immunotherapy (*Lei et al., 2024*; *Catanzaro et al., 2024*). Thus, ferroptosis may serve as a new strategy for increasing the efficacy of immunotherapy combined with Cis. It is well established that p53, a classic tumor suppressor protein, can regulate ferroptosis via multiple mechanisms (*Liu and Gu, 2022*; *Jiang et al., 2015*). As a traditional transcription factor (TF), p53 transcriptionally regulates several metabolic or ferroptotic targets, such as *TIGAR, GLS2*, and *ALOX12*, to mediate ferroptosis (*Suzuki et al., 2010*; *Chu et al., 2019*; *Wang et al., 2016*). However, inactivation of p53 tumor suppressor function is very common and plays an important role in the progression of most cancers (*Wang and Attardi, 2022*). Among patients with GC, those with activated p53 status have longer overall survival (OS) than those with p53 loss of function (*Cristescu et al., 2015*). Recently, studies have shown that activating p53 can increase chemotherapy efficacy by promoting ferroptosis (*Shao et al., 2024*; *Geng et al., 2024*). Therefore, activating p53 function in GC is a highly desired feature of ferroptosis-based antitumor therapy.

Nucleolar and coiled-body phosphoprotein 1 (NOLC1), also called NOPP140, was originally identified as a nuclear localization signal binding protein (NLS) and functions as a molecular chaperone that shuttles between the cytoplasm and nucleus (*Thomas Meier and Blobel, 1992*; *Isaac et al., 1998*). NOLC1 is upregulated in most cancers and promotes non-small cell lung cancer (NSCLC) resistant to multiple drugs (*Huang et al., 2018*). Additionally, NOLC1 interacts with the MDM2 promoter at the p53 binding site, and its functional regulation by p53 was confirmed through an RNAi-mediated synthetic interaction screen (*Krastev et al., 2011*; *Hwang et al., 2009*). However, the specific expression level and functions of NOLC1 in GC remain unclear.

Here, we found that NOLC1 is upregulated in GC tumors and GC-resistant cells. High expression of NOLC1 promotes GC Cis resistance by suppressing ferroptosis. Further studies revealed that NOLC1 interacts with the p53 DNA-binding domain (DBD) and suppresses p53 nuclear accumulation, thus inhibiting p53 transcription-mediated ferroptosis. Eventually, the knockdown of NOLC1 activated the ferroptosis-mediated ICD and significantly improved the efficacy of low-dose anti-PD-1 plus Cis without side effects. Collectively, this study identified a novel network mediated by NOLC1 that regulates p53-mediated ferroptosis in GC and identified a new target for GC immunotherapy in combination with Cis.

## Results

### High expression of NOLC1 is associated with poor clinical outcomes in GC

Recent studies have demonstrated that NOLC1 is upregulated in most cancers (*Chen et al., 2021b*; *Kong et al., 2021*). However, the expression of NOLC1 in GC is still unknown. To address this gap in

knowledge, we collected GC patients' tumor and near-tumor tissues to measure NOLC1 expression level. As shown in *Figure 1A*, *Figure 1—figure supplement 1A*, immunohistochemical (IHC) staining demonstrated that NOLC1 protein level was much higher in GC tissues than in near-tumor tissues. Compared with that of normal tissues, the average optical density score was notably greater in GC tissues (*Figure 1B*). Consistently, as shown in *Figure 1C and D*, western blotting (WB) results also revealed the same trend: NOLC1 was highly expressed in GC tissues. Analysis of the TCGA database also revealed that NOLC1 is upregulated in GC tissues (*Figure 1E and F*). We next used the Kaplan-Meier plotter database to assess whether high expression of NOLC1 influenced GC patient outcomes (*Győrffy, 2024*). Higher NOLC1 expression level was related to shorter OS, time to first progression (FP), and post-progression survival in patients with GC (*Figure 1G–I*). Taken together, all these results suggest that NOLC1 protein levels are increased in GC and that NOLC1 may promote the progression of GC.

Since NOLC1 is highly expressed and promotes resistance to multiple drugs in NSCLC (*Huang et al., 2018*), we further detected whether NOLC1 is associated with GC resistance. To investigate the expression of NOLC1 in Cis-resistant GC, a Cis-resistant MGC-803 (MGC-803-CR) cell line was established. CCK-8, colony formation, and Annexin V-APC/7-AAD apoptosis assays revealed that MGC-803-CR cells were less sensitive to Cis than MGC-803 cells (*Figure 1—figure supplement 2A-E*). Next, the expression of NOLC1 in MGC-803-CR cells was detected. WB showed that NOLC1 is highly expressed in MGC-803-CR cells (*Figure 1J*, *Figure 1—figure supplement 3A*). The cleaved PARP protein level was significantly lower than that in MGC-803 cells (*Figure 1—figure supplement 3B*), further supporting that MGC-803-CR cells are resistant to Cis. Moreover, immunofluorescence (IF) analysis revealed that NOLC1 was upregulated in both the cytoplasm and nucleolus (*Figure 1K*, *Figure 1—figure supplement 3C*). These data indicate that NOLC1 is highly expressed in the Cis resistance cell line. In addition, we used mRNA-seq to analyze the specific biological pathway alterations in MGC-803-CR cells. As shown in *Figure 1—figure supplement 3D and E*, the expression level of components of the Hippo signaling pathway, PI3K-AKT pathway, p53 signaling pathway, etc. significantly differed between MGC-803-CR cells and MGC-803 cells.

## NOLC1 knockdown increases the sensitivity of GC to Cis in vitro and in vivo

To investigate the functions of NOLC1 in Cis resistance GC, lentiviral shRNA constructs (sh*NOLC1*#1, #2, or #3) were used to knock down endogenous *NOLC1* gene expression in GC cell lines, MGC-803 and MKN-45 cells, respectively (*Figure 2—figure supplement 1A–D*). CCK-8 assay revealed that NOLC1 silencing significantly decreased the viability of MGC-803 and MKN-45 cells upon Cis treatment (*Figure 2A*). Meanwhile, overexpression (OE) of NOLC1 promoted Cis resistance in MGC-803 cells (*Figure 2—figure supplement 1E*, *Figure 2—figure supplement 2A*). The colony formation assay results also revealed that knockdown of NOLC1 could significantly decrease the numbers of colonies after Cis treatment (*Figure 2B and C*). Moreover, the Annexin V-APC/7-AAD apoptosis assay revealed that Cis induced cell death to a greater degree in the NOLC1-knockdown group than that in the control group (*Figure 2D and E*), suggesting that NOLC1 enhances the Cis resistance in GC cells.

Cis exerts its cytotoxic effects by binding to DNA to form DNA-platinum adducts and destroy DNA (*Shu et al., 2016*). The toxic effects of Cis on DNA were further evaluated via IF staining of γ-H2AX (Ser-139) and comet assays. The IF results revealed increased aggregation of γ-H2AX foci in the knockdown group after treatment with Cis (*Figure 2F and G*). Comet assays involving electrophoresis were performed and revealed that negligible DNA fragments migrated out of the nucleus in the negative control (NC) group treated with phosphate-buffered saline (PBS), while DNA tails were observed outside the nucleus after treatment with Cis (*Figure 2H*). In particular, a more pronounced tailing shape of DNA strands was found in the knockdown group, especially after treatment with Cis, indicating that NOLC1 knockdown enhances Cis-induced DNA damage. Moreover, the WB results revealed that the protein levels of cleaved caspase-3 and cleaved PARP were increased after treatment with Cis, especially in the knockdown group, which presented a greater degree of increasing accompanied by decreased expression of PARP and caspase-3 (*Figure 2I*, *Figure 2—figure supplement 2B and C*), suggesting that Cis induced more severe DNA damage in the NOLC1-knockdown group. Taken together, these findings indicate that NOLC1 knockdown enhances Cis cytotoxicity in GC cells.

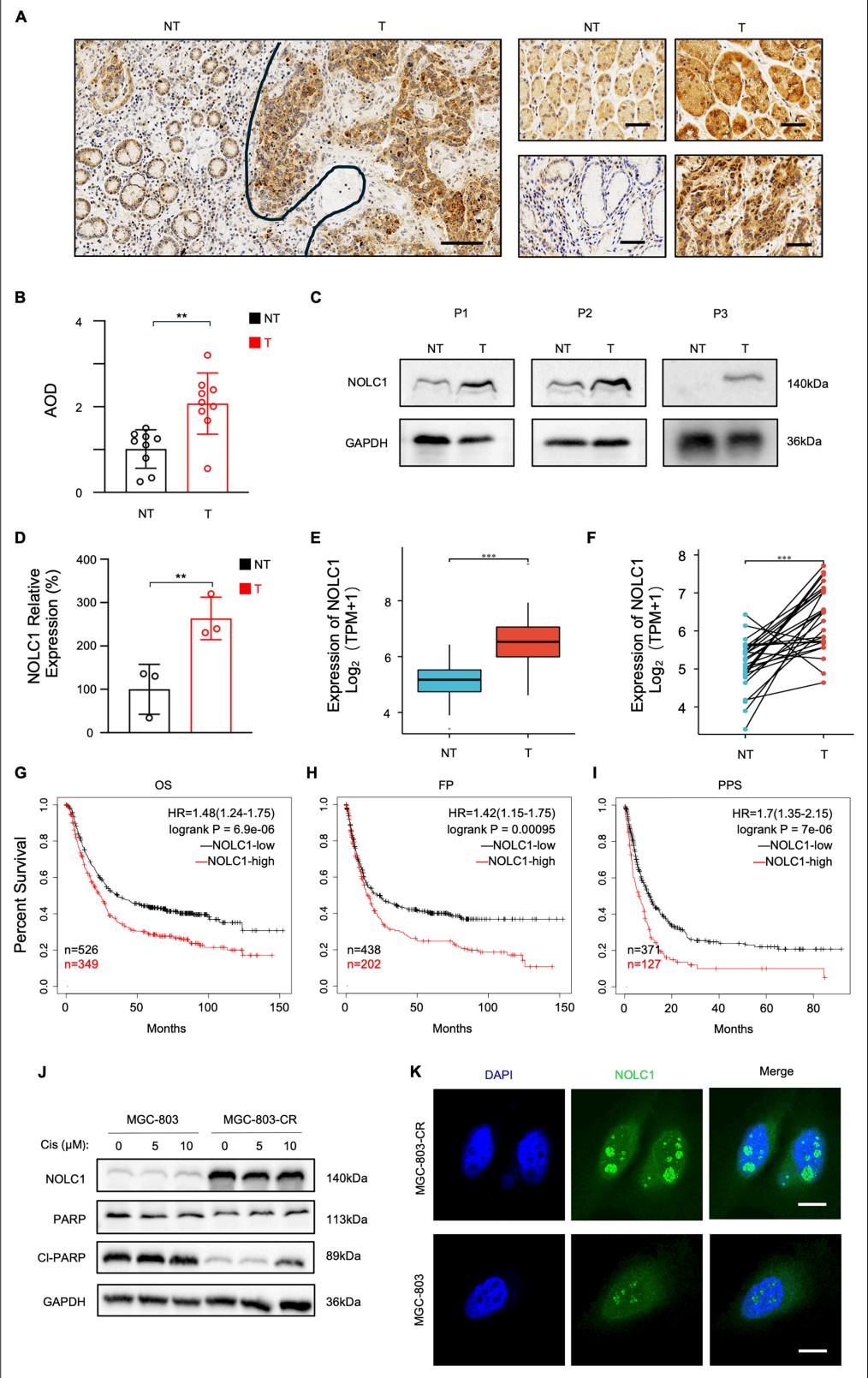

**Figure 1.** NOLC1 was upregulated in gastric cancer (GC) and a cisplatin (Cis)-resistant GC cell line. (**A, B**) Immunohistochemistry (IHC) results of NOLC1 expression in gastric tumor (T) tissues and near-tumor (NT) tissues. (**A**) Representative images (n=3); scale bar = 50 μm. (**B**) Average optical density (AOD) of NOLC1 according to the IHC images (n=9). (**C, D**) Western blotting (WB) results of NOLC1 protein levels in gastric tumor (T) tissues and

*Figure 1 continued on next page*

*Figure 1 continued*

NT tissues. (**C**) Representative images. (**D**) Relative expression of NOLC1 (n=3). (**E, F**) Gene expression level of NOLC1 in TCGA GC tumor and matched TCGA normal stomach tissues. (**E**) Unpaired sample; (**F**) paired sample. (**G–I**) High NOLC1 expression predicts poor survival in GC patients. (**G**) Overall survival (OS); patients were divided into a NOLC1-high group (n=349) and a NOLC1-low group (n=526). (**H**) Time to first progression (FP); patients were divided into a NOLC1-high group (n=202) and a NOLC1-low group (n=438). (**I**) Post-progression survival (PPS); patients were divided into a NOLC1-high group (n=127) and a NOLC1-low group (n=371). (**J**) WB results of NOLC1, PARP, and cleaved PARP in MGC-803 and MGC-803-CR cell lines. (**K**) Immunofluorescence (IF) images of NOLC1 in MGC-803 and MGC-803-CR cell lines; scale bar = 15 μm. The data are presented as the means ± SDs. **p<0.01; ***p<0.001.

The online version of this article includes the following source data and figure supplement(s) for figure 1:

**Source data 1.** Original western blots for *Figure 1C and J*, indicating the relevant bands.

**Source data 2.** Original files for western blot displayed in *Figure 1C and J*.

**Figure supplement 1.** NOLC1 was highly expressed in gastric cancer (GC) tissues.

**Figure supplement 2.** Cisplatin (Cis)-resistant gastric cancer (GC) cell line was successfully constructed.

**Figure supplement 3.** NOLC1 was highly expressed in CR cells.

To determine whether NOLC1 affects GC Cis resistance in vivo, we constructed xenograft GC models in BALB/c-nu mice. The mice were randomly divided into two groups and subcutaneously injected with MGC-803 cells (sh*NC* or sh*NOLC1*). When the tumor volume reached about 100 mm$^3$, each group was randomly divided into two subgroups (PBS and Cis), for a total of four groups. Cis or PBS was then administered to the mice via intraperitoneal injection twice per week for a total of six times (*Figure 3A*). The results showed that Cis treatment effectively slowed the growth of the xenografts. The NOLC1-knockdown groups were more sensitive to Cis (*Figure 3B–E*). Next, to further compare the therapeutic effects, we performed hematoxylin and eosin (H&E) staining and IHC staining to detect cleaved caspase-3 and Ki-67 expression in the tumors. H&E staining revealed that the NOLC1-knockdown groups presented a larger necrotic area in the tumor tissues after Cis treatment (*Figure 3F and G*). The IHC results revealed that Cis decreased Ki-67 protein levels, and the NOLC1-knockdown groups presented lower levels than did the NC group. The cleaved caspase-3 level was increased after Cis treatment, and in the knockdown groups, the cleaved caspase-3 level was greater than that in the NC groups. Collectively, all these data indicate that NOLC1 knockdown increased Cis sensitivity in gastric xenografts in vivo.

## NOLC1 suppresses Cis-induced ferroptosis

Cis can induce apoptosis to kill cancers (*Romani, 2022*). Also, other functions of Cis, such as inducing ferroptosis, which plays a pivotal role in Cis resistance, have also been reported (*Lin et al., 2023*; *Li et al., 2022*). To determine which type of RCD is primarily blocked by NOLC1, we used transmission electron microscopy (TEM) to observe specific morphological changes in the MGC-803 and MKN-45 cells. The results revealed that cells with NOLC1 knockdown exhibited shrunken mitochondria and increased membrane density. After Cis treatment, the mitochondrial membrane was severely disrupted, and the mitochondrial cristae disappeared in the knockdown group (*Figure 4A*, *Figure 4—figure supplement 1A*). These are the classic characteristic morphological features of ferroptosis (*Ouyang et al., 2022*). Moreover, the erastin (a ferroptosis activator) induced cell death was also suppressed by NOLC1 (*Figure 4—figure supplement 1B*). Fer-1 (a ferroptosis inhibitor) could rescue the sensitization effect of NOLC1 knockdown (*Figure 4—figure supplement 1C*). These findings indicate that NOLC1 might strongly inhibit ferroptosis to promote Cis resistance. During ferroptosis, ferroptotic cell lipid membrane structure is disrupted. We further examined specific ferroptotic cell lipid membrane morphological changes in response to Cis after NOLC1 knockdown. Lactate dehydrogenase (LDH) release assays revealed that after treatment with Cis, higher levels of LDH were detected in the knockdown groups (*Figure 4B*, *Figure 4—figure supplement 1D*), indicating more severe destruction of the membrane structure in the knockdown group. Meanwhile, the high level of LDH release could be blocked by Fer-1 in knockdown groups, further reflecting that NOLC1 promotes Cis resistance by suppressing ferroptosis (*Figure 4—figure supplement 1E*). Moreover, the JC-1 probe results indicated that the mitochondrial membrane potential decreased in the NOLC1-knockdown

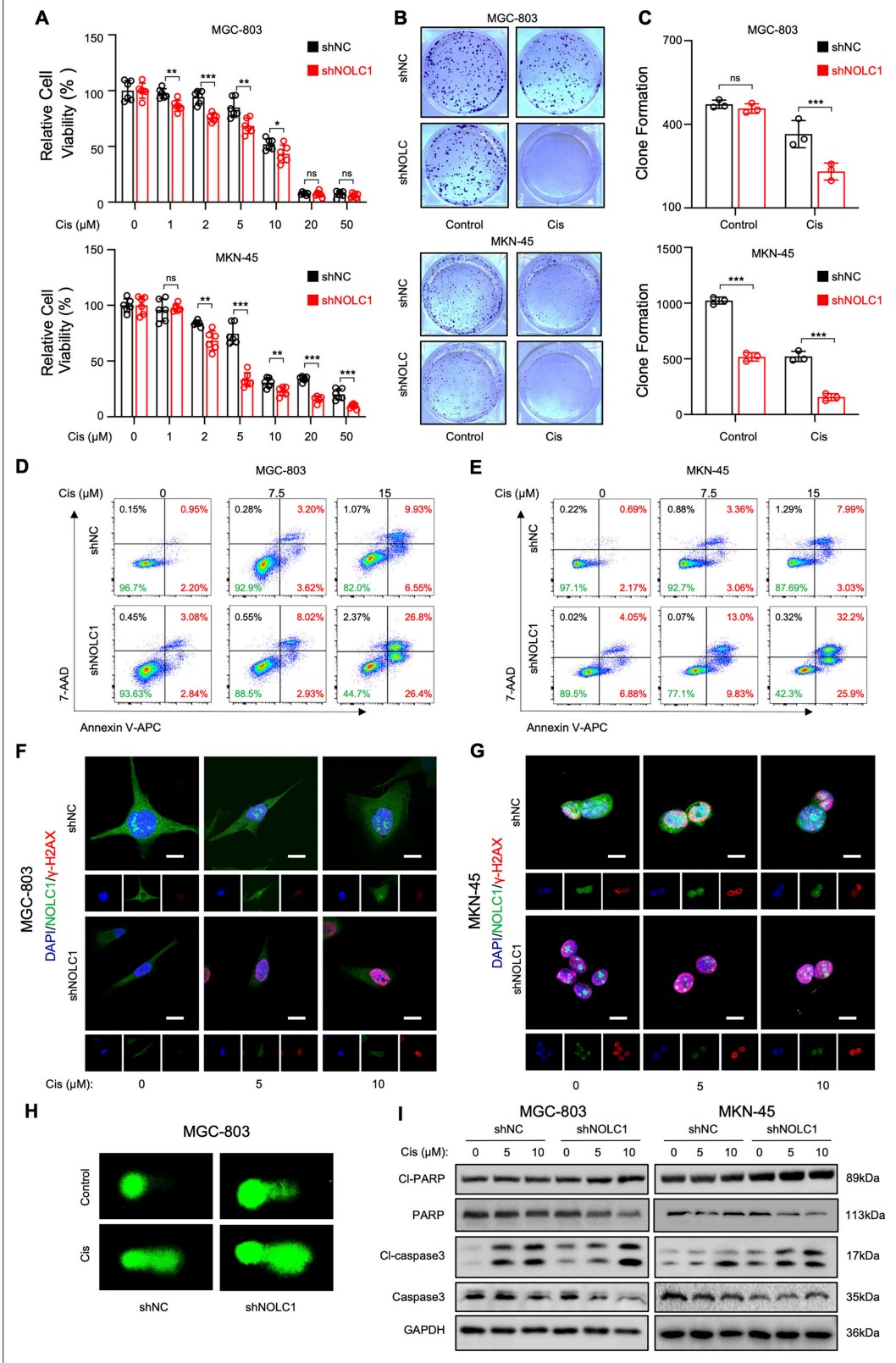

**Figure 2.** NOLC1 promoted cisplatin (Cis) resistance in gastric cancer (GC). (**A**) CCK-8 assay of GC cells transduced with sh*NC* or sh*NOLC1* lentivirus and treated with different concentrations of Cis (n=6). (**B, C**) Colony formation assay of GC cells transduced with sh*NC* or sh*NOLC1* lentivirus and treated with phosphate-buffered saline (PBS) or Cis (15 μM). (**B**) Representative images of the colony formation assay. (**C**) Quantitative analysis of colony

*Figure 2 continued on next page*

*Figure 2 continued*

formations assay (n=3). (**D, E**) Annexin V-APC and 7-AAD staining of GC cells transduced with sh*NC* or sh*NOLC1* lentivirus and treated with the indicated concentrations of Cis, as analyzed via fluorescence-activated cell sorting (FACS); (**D**) MGC-803 cells and (**E**) MKN-45 cells. (**F, G**) Immunofluorescence staining assays of γ-H$_2$AX in GC cells transduced with sh*NC* or sh*NOLC1* lentivirus and treated with the indicated concentrations of Cis; (**F**) MGC-803 cells; scale bar = 10 μm; (**G**) MKN-45 cells; scale bar = 5 μm. (**H**) Comet assay of MGC-803 cells transduced with sh*NC* or sh*NOLC1* lentivirus and treated with PBS or Cis (15 μM). (**I**) Western blotting (WB) results of cleaved PARP, PARP, cleaved caspase-3, and caspase-3 in GC cells transduced with sh*NC* or sh*NOLC1* lentivirus and treated with indicated concentrations of Cis. The data are presented as the means ± SDs. ns, nonsignificant; *p<0.05; **p<0.01; ***p<0.001.

The online version of this article includes the following source data and figure supplement(s) for figure 2:

**Source data 1.** Original western blots for *Figure 2I*, indicating the relevant bands.

**Source data 2.** Original files for western blot displayed in *Figure 2I*.

**Figure supplement 1.** Knockdown or overexpression of NOLC1 in gastric cancer (GC) cell lines.

**Figure supplement 1—source data 1.** Original western blots for *Figure 2—figure supplement 1C-E*, indicating the relevant bands.

**Figure supplement 1—source data 2.** Original files for western blot displayed in *Figure 2—figure supplement 1C-E*.

**Figure supplement 2.** NOLC1 promoted cisplatin (Cis) resistance in gastric cancer (GC) cells.

---

group after Cis treatment, indicating that the mitochondrial membrane was also disrupted (*Figure 4C*, *Figure 4—figure supplement 1F and G*). These data revealed that NOLC1-knockdown GC cells displayed classic ferroptosis-related morphological features.

Disruption of the cell lipid membrane during ferroptosis is caused by lipid peroxides (*Jiang et al., 2021*). Therefore, we further detected the reactive oxygen species (ROS) level after NOLC1 knockdown using 2,7-dichlorodihydrofluorescein diacetate (DCFH-DA) staining and analyzed by fluorescence-activated cell sorting (FACS) and confocal microscopy. Compared with the negligible ROS accumulation of the NC groups, the knockdown groups presented a greater level of ROS accumulation, especially when treated with Cis (*Figure 4D*, *Figure 4—figure supplement 1H and I*). MitoPeDPP, which accumulates in the inner mitochondrial membrane, can be oxidized by lipid peroxides and release strong fluorescence. Using MitoPeDPP probes, we found that, in the NOLC1-knockdown groups, the rate of mitochondrial-specific ROS accumulation was significantly higher than that in the NC group at the same Cis treatment time (*Figure 4E*, *Figure 4—figure supplement 1J*). Also, a greater level of fluorescence accumulation in the mitochondria was observed after treatment with Cis in the knockdown groups (*Figure 4F*). Moreover, the H$_2$O$_2$ content increased after Cis treatment, and the H$_2$O$_2$ content in the knockdown groups was much greater than that in the NC groups (*Figure 4G*). These data indicated that lipid ROS significantly accumulated in the NOLC1-knockdown group. In line with the above data, the key protein involved in ferroptosis, GPX4, was obviously downregulated in the NOLC1-knockdown group both in vitro and in vivo (*Figure 4H*, *Figure 4—figure supplement 2A and B*). Moreover, FSP1 was decreased and ACSL4 was increased in the knockdown group, reflecting that NOLC1 could effectively suppress ferroptosis. Taken together, these data show that NOLC1 primarily inhibits Cis-induced ferroptosis to promote GC resistance.

## NOLC1 binds to p53 and blocks p53 translocation

Next, we further investigated the molecular mechanism of which NOLC1 promotes resistance in GC. Recent research has shown that the function of NOLC1 is associated with p53 (*Krastev et al., 2011*; *Hwang et al., 2009*). In addition, via mRNA-seq, we found that the p53 signal pathway was obviously altered in MGC-803-CR cells (*Figure 1—figure supplement 3E*). Thus, we assumed that NOLC1-mediated suppression of ferroptosis promotes GC resistance via regulating p53.

To verify our hypothesis, the p53 protein level was measured, and the results indicated that the p53 protein level increased when the cells were treated with Cis, further confirming that p53 is upregulated in stressed cells (*Figure 5A*, *Figure 5—figure supplement 1A*). However, in the knockdown group, the p53 level was lower than that in the NC group and was not associated with cell death trends. BAX is a pro-apoptotic protein whose transcription is mediated by p53 (*Seo et al., 2022*). BCL-2 is

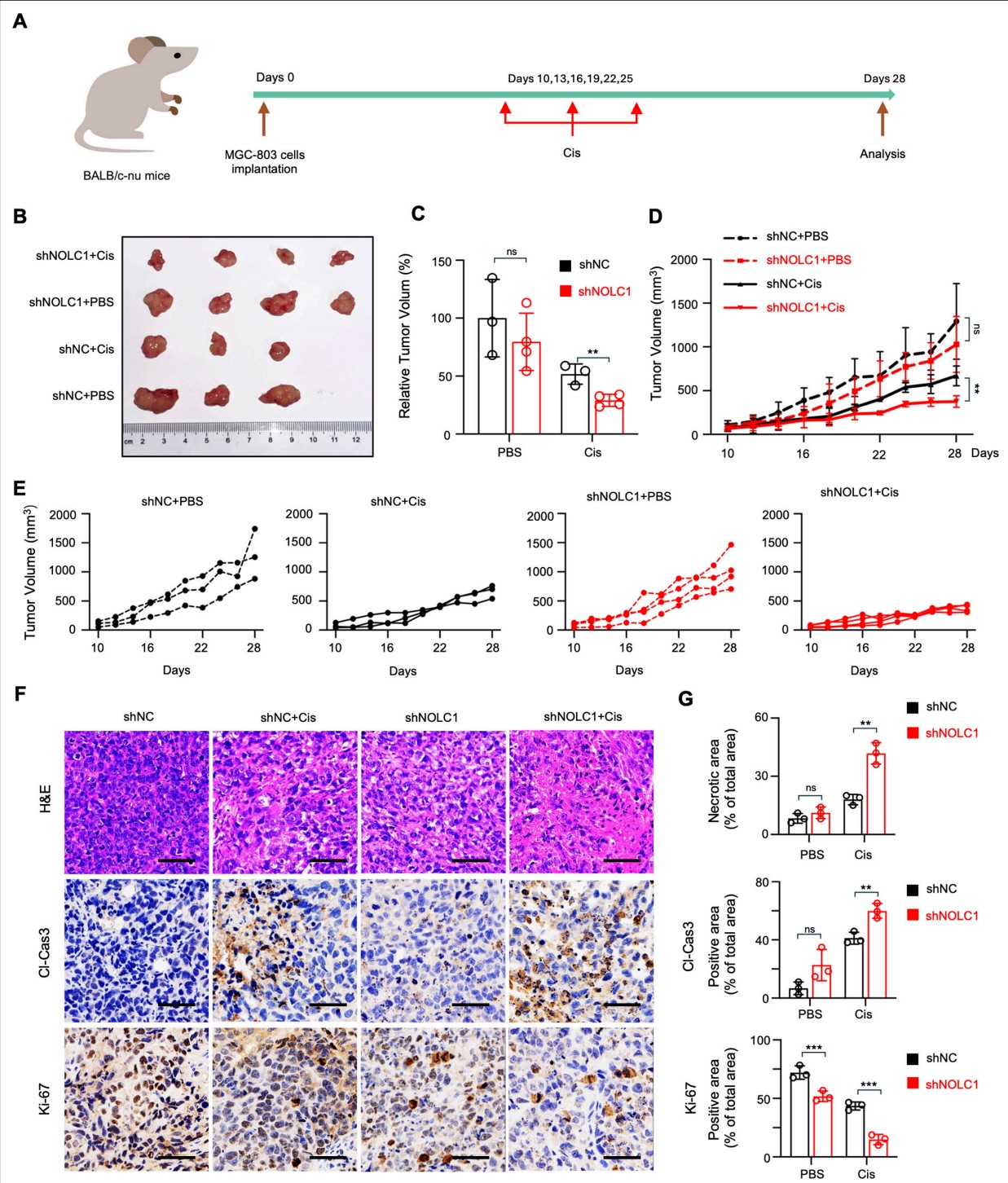

**Figure 3.** NOLC1 promoted cisplatin (Cis) resistance in vivo. (**A**) Schematic representation showing the treatment of the subcutaneous xenograft tumor models. (**B**) Photographs of the corresponding tumors after indicated treatments. (**C**) Relative tumor volume after different treatments. (**D, E**) Tumor growth curves after indicated treatments. (**F**) Hematoxylin and eosin (H&E) staining and immunohistochemical (IHC) analysis of cleaved caspase-3 and Ki-67; scale bar = 50 μm. (**G**) Quantification analysis of the necrotic area and cleaved caspase-3 and Ki-67 expression in tumor tissue from mice given the indicated treatments (n=3). The data are presented as the means ± SDs. ns, nonsignificant; *p<0.05; **p<0.01; ***p<0.001.

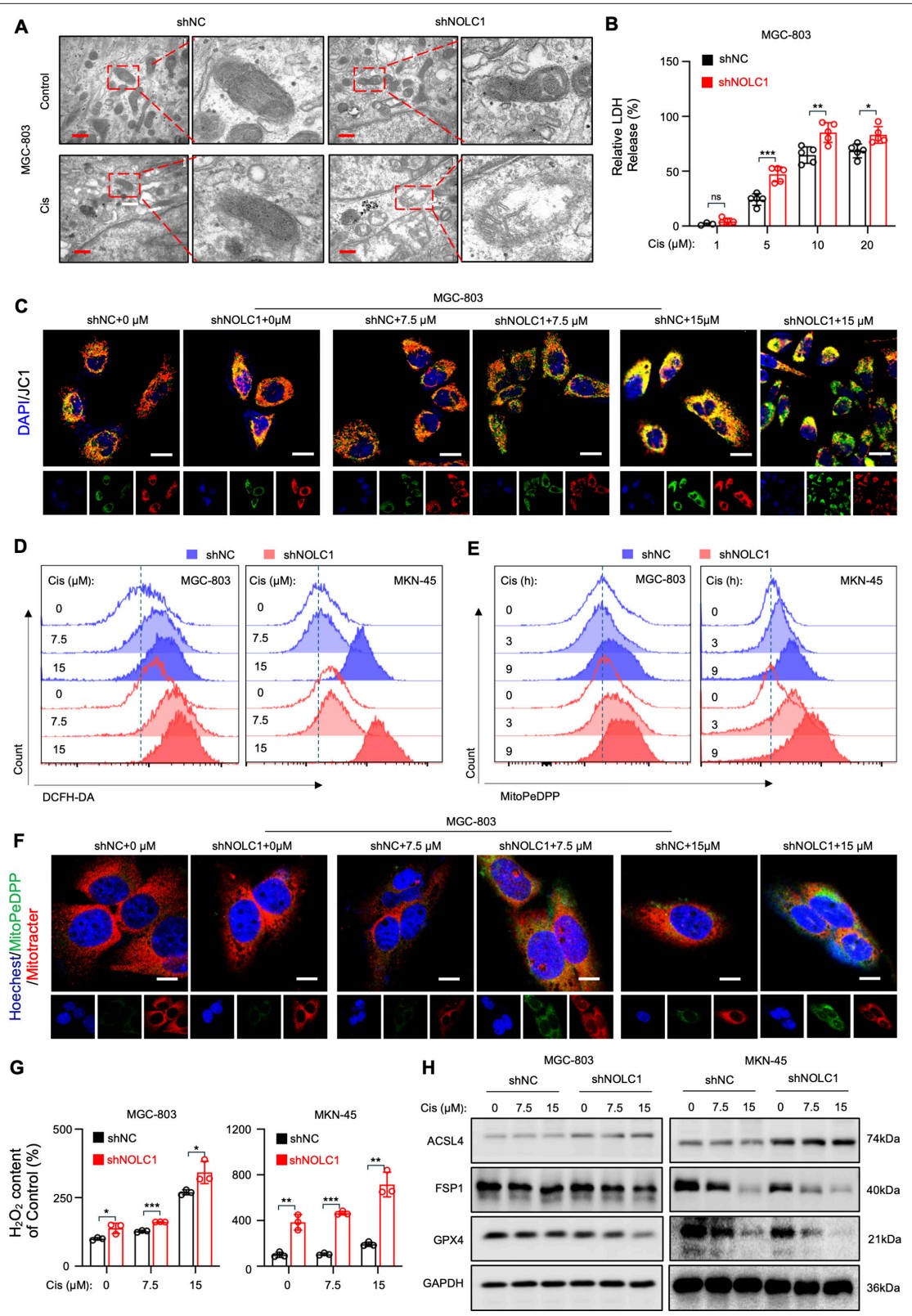

**Figure 4.** NOLC1 silence rendered gastric cancer (GC) susceptible to ferroptosis. (**A**) Transmission electron microscopy (TEM) images of MGC-803 cells transduced with sh*NC* or sh*NOLC1* lentivirus and treated with phosphate-buffered saline (PBS) or cisplatin (Cis) (15 μM); scale bar = 400 nm. (**B**) Lactate dehydrogenase (LDH) release analysis of MGC-803 cells transduced with sh*NC* or sh*NOLC1* lentivirus and treated with indicated concentrations of Cis (n=5). (**C**) Representative JC-1 fluorescence images of MGC-803 cells transduced with sh*NC* or sh*NOLC1* lentivirus and treated with indicated

*Figure 4 continued on next page*

*Figure 4 continued*

concentrations of Cis; scale bar = 20 µm. (**D**) Fluorescence intensity of 2,7-dichlorodihydrofluorescein diacetate (DCFH-DA) measured by fluorescence-activated cell sorting (FACS) in GC cells transduced with sh*NC* or sh*NOLC1* lentivirus and treated with indicated concentrations of Cis. (**E**) Fluorescence intensity of MitoPeDPP measured by FACS in GC cells transduced with sh*NC* or sh*NOLC1* lentivirus and treated with Cis (15 µM) for different times. (**F**) Representative MitoPeDPP fluorescence images of MGC-803 cells transduced with sh*NC* or sh*NOLC1* lentivirus and treated with indicated concentrations of Cis; scale bar = 10 µm. (**G**) Relative $H_2O_2$ content of GC cells transduced with sh*NC* or sh*NOLC1* lentivirus and treated with indicated concentrations of Cis (n=3). (**H**) Western blotting (WB) result of ACSL4, FSP1, and GPX4 protein levels in GC cells transduced with sh*NC* lentivirus or sh*NOLC1* lentivirus and treated with different concentrations of Cis. The data are presented as the means ± SDs. ns, nonsignificant; *p<0.05; **p<0.01; ***p<0.001.

The online version of this article includes the following source data and figure supplement(s) for figure 4:

**Source data 1.** Original western blots for *Figure 4H*, indicating the relevant bands.

**Source data 2.** Original files for western blot displayed in *Figure 4H*.

**Figure supplement 1.** NOLC1 silence rendered gastric cancer (GC) cells susceptible to ferroptosis.

**Figure supplement 2.** NOLC1 silence rendered gastric cancer (GC) cells susceptible to ferroptosis.

an anti-apoptotic protein that is mediated by p53 via protein interaction in the cytoplasm. Interestingly, the increase of BAX level after NOLC1 knockdown was out of line with the p53 protein level trend but in line with the cell death trend, and the BCL-2 level was in line with p53 level (*Figure 5A*, *Figure 5—figure supplement 1B*). Additionally, according to the IHC results, p53 expression was also decreased in the knockdown group in vivo (*Figure 5—figure supplement 1C*). Thus, we assume that after NOLC1 is knocked down, the transcriptional functions of p53 increased, but the protein level decreased. To validate our hypothesis, we detected the transcriptional activity levels of p53 and downstream genes of p53 (*CDKN1A, BAX, FAS,* and *PTEN*) expression. Dual-luciferase assay showed that p53 transcription is activated by Cis, indicating that p53 is activated when cells are stressed. Knocking down NOLC1 significantly increased p53 activity in both MGC-803 and MKN-45 cells (*Figure 5B*, *Figure 5—figure supplement 1D*). Meanwhile, NOLC1 OE in HEK-293T cells obviously decreased p53 transcriptional activity in a dose-dependent manner (*Figure 5C*). Additionally, the activity of the NC reporter and that of the mut p53 reporter did not significantly differ (*Figure 5—figure supplement 1E and F*). Real-time quantitative reverse transcription polymerase chain reaction (RT-qPCR) results revealed that *CDKN1A, BAX, FAS,* and *PTEN* mRNA levels were increased in the knockdown group both in vitro and in vivo (*Figure 5—figure supplement 1G and H*). Collectively, these results indicate that NOLC1 can decrease the transcriptional activity of p53.

Considering that, as a molecular chaperone, NOLC1 can interact with other proteins, such as TRF2 (*Yuan et al., 2017*), to regulate their functions, we assume that NOLC1 regulates p53 functions via interaction with p53. Thus, the docking studies were performed. The results showed that NOLC1 can interact with p53 (*Figure 5D*). The binding energy of NOLC1 to p53 was –236.13 kcal/mol, and the binding sites of p53 are all in the DNA-binding domain (DBD), which is responsible for transcription (*Table 1*). In addition, the surface of the p53 protein matched well with that of the NOLC1 protein, which promoted stable binding (*Figure 5E and F*). To verify the potential interaction between NOLC1 and p53, we carried out coimmunoprecipitation (Co-IP) assays in HEK-293T cells. As expected, immunoprecipitation of p53-HA coprecipitated with NOLC1-Flag demonstrated that NOLC1 can interact with p53 (*Figure 5G*). The IF results also revealed that NOLC1 colocalized with p53 in both HEK-293T and MGC-803 cells (*Figure 5—figure supplement 2A*). The nuclear accumulation of p53 is crucial for its transcriptional function. To detect whether NOLC1 affects the p53 nuclear accumulation upon Cis treatment, IF was applied to detect the p53 location after Cis treatment in HEK-293T cells. Under normal conditions, NOLC1 and p53 colocalized in the cytoplasm, and p53 accumulated in the nucleus after Cis treatment. After OE of NOLC1, the nuclear level of p53 significantly decreased under Cis treatment, and as NOLC1 expression increased, the nuclear/cytoplasmic ratio of p53 gradually decreased further (*Figure 5H*, *Figure 5—figure supplement 2B*). In addition, knockdown of NOLC1 significantly increased the nuclear/cytoplasmic ratio of p53 under normal conditions (*Figure 5—figure supplement 2C*). The WB results revealed similar trends in the GC cells (*Figure 5I*). Additionally, IHC staining revealed that the ratio of p53 nuclear positive cells was significantly increased (*Figure 5—figure supplement 1C*). These data indicate that NOLC1 inhibits p53 nuclear accumulation.

The full-length p53 protein has three conserved functional domains: a transactivation domain, a DBD, and a tetramerization domain (*Senitzki et al., 2021*; *Figure 5J*). To determine which domain of

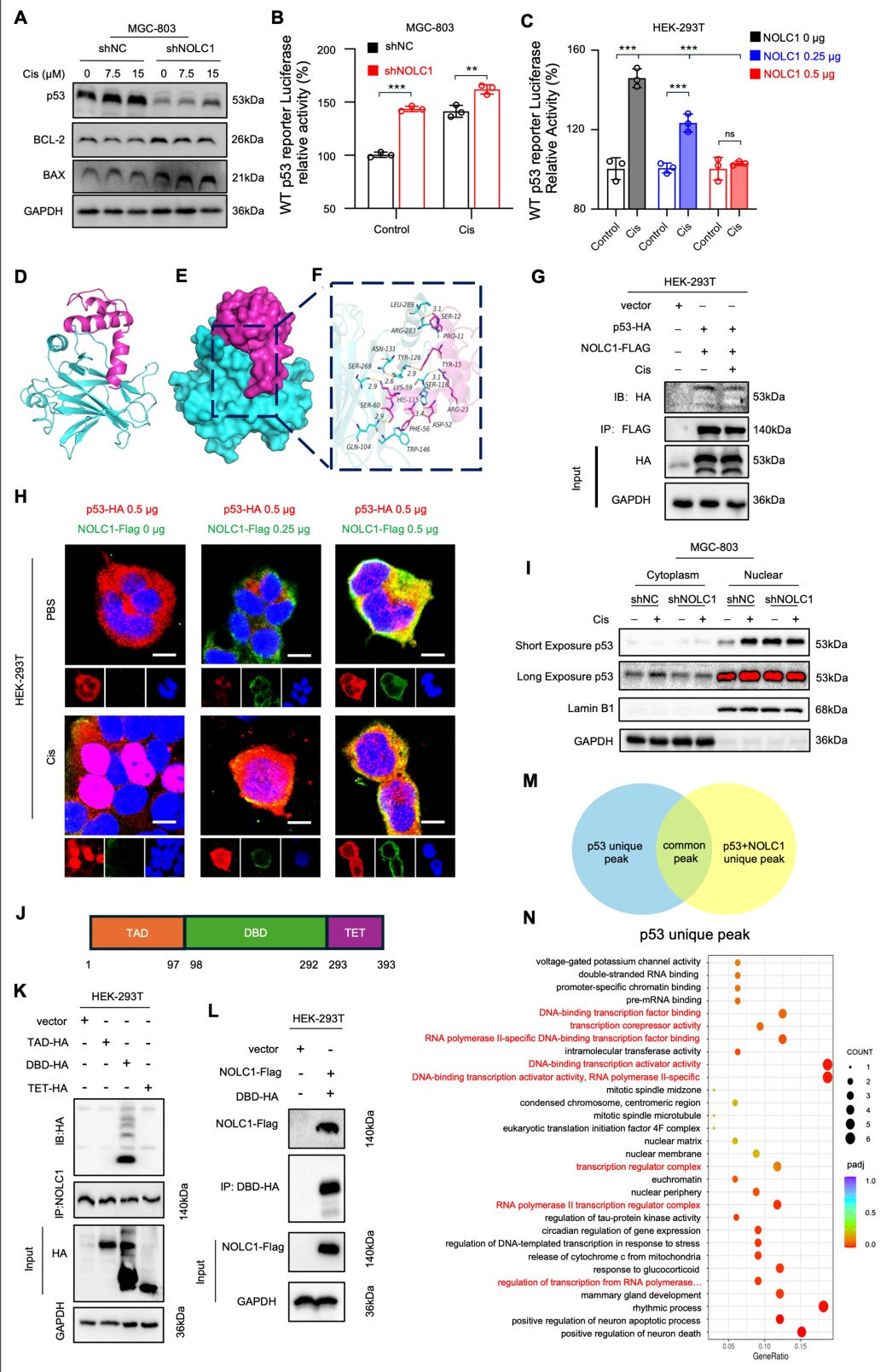

**Figure 5.** NOLC1 decreased p53 transcriptional activity. (**A**) Western blotting (WB) analysis of p53, BCL-2, and BAX in MGC-803 cells transduced with shNC or shNOLC1 lentivirus and treated with indicated concentrations of cisplatin (Cis). (**B**, **C**) Luciferase assay of p53 transcription activity (**B**) in MGC-803 cells transduced with sh*NC* or sh*NOLC1* lentivirus and treated with phosphate-buffered saline (PBS) or Cis (15 μM) and (**C**) in HEK-293T cells

*Figure 5 continued on next page*

*Figure 5 continued*

transfected with 0.00, 0.25, or 0.50 µg NOLC1 plasmid and treated with PBS or Cis (15 µM). (**D–F**) Docking analysis of NOLC1 and p53; (**D**) The backbones of the proteins were shown in tubes and colored red (NOLC1) and cyan (p53); (**E**) The NOLC1 and p53 proteins were shown on the surface; (**F**) The detailed binding mode of NOLC1 with p53. The yellow dash represented the hydrogen bond. (**G**) Coimmunoprecipation (Co-IP) assay of NOLC1-FLAG with p53-HA. (**H**) Representative NOLC1-Flag and p53-HA fluorescence images of HEK-293T transfected with indicated amount of NOLC1-Flag and p53-HA plasmids and treated with PBS or Cis (15 µM); scale bar = 10 µm. (**I**) WB analysis of p53 protein levels in the cytoplasm or nucleus of MGC-803 cells transduced with sh*NC* or sh*NOLC1* lentivirus and treated with PBS or Cis (15 µM). (**J**) Schematic map of the p53 functional domain. (**K**) Co-IP assay of NOLC1 with different p53 functional domains. (**L**) Co-IP assay of p53 DNA-binding domain (DBD)-HA with NOLC1-Flag. (**M**) Schematic map of the chromatin immunoprecipitation (ChIP-seq) analysis. (**N**) GEO analysis of the unique p53 peak identified via ChIP-seq. The data are presented as the means ± SDs. ns, nonsignificant; \*\*p<0.01; \*\*\*p<0.001.

The online version of this article includes the following source data and figure supplement(s) for figure 5:

**Source data 1.** Original western blots for *Figure 5A, G, I, K, and L*, indicating the relevant bands.

**Source data 2.** Original files for western blot displayed in *Figure 5A, G, I, K, and L*.

**Figure supplement 1.** NOLC1 inhibited p53 transcriptional activity.

**Figure supplement 2.** NOLC1 combined with p53 and inhibited p53 nuclear accumulation.

---

p53 is responsible for its interaction with NOLC1, three functional domain constructs were generated to perform Co-IP assays. Consistent with the docking results (*Table 1*), Co-IP assays revealed that only the DBD can interact with NOLC1 (*Figure 5K*). Meanwhile, the DBD-HA could coprecipitate NOLC1-Flag too (*Figure 5L*). Considering that the interaction between NOLC1 and p53 DBD might block p53 binding to DNA, we performed chromatin immunoprecipitation (ChIP-seq) to determine whether this interaction could inhibit p53 binding to specific DNA sequences. We transfected HEK-293T cells with a p53 full-length plasmid or the p53 full-length plus NOLC1 plasmid (*Figure 5M*). Then, p53 immunoprecipitation confirmed the binding of p53 to the DNA sequence. As shown in *Figure 5N*, the p53 group could uniquely bind to DNA-binding transcription activator activity peaks, DNA-binding TF binding peaks, etc. These findings demonstrated that NOLC1 could inhibit p53 binding to transcription-related DNA sequences. Furthermore, the data indicated that NOLC1 binds to the p53 DBD, not only to mediate p53 nuclear accumulation but also to block p53 binding to DNA, eventually blocking p53-mediated ferroptosis.

Taken together, the above data showed that NOLC1 inhibits p53 nuclear accumulation stimulated by Cis treatment and transcriptional activity by interacting with the p53 DBD.

## NOLC1 inhibits p53 degradation via disruption of the MDM2-p53 feedback loop

Although we have shown that NOLC1 decreases p53 transcriptional functions, it is still unclear why the p53 protein level decreases after NOLC1 is knocked down (*Figure 5A*). RT-qPCR results showed that there was no significant difference in the *TP53* mRNA level between the NC and knockdown groups (*Figure 6A*). Thus, we speculated that NOLC1 increases the p53 protein level by inhibiting its degradation. The E3 ubiquitin ligase MDM2, encoded by a p53-responsive gene, is the master antagonist of p53 by promoting p53 ubiquitination and proteasomal degradation, forming a negative feedback loop (*Brooks and Gu, 2006*). On the basis of the above data, we hypothesized that NOLC1 inhibits p53 transcription-mediated *MDM2*, thereby reducing p53 ubiquitination and degradation. Therefore, we next detected the *MDM2* mRNA and protein levels. As shown in *Figure 6B–D*, the

**Table 1.** The docking results of the two target proteins – NOLC1 and p53.

| Protein 1 | Protein 2 | Binding energy (kcal/mol) | Contact sites (protein 1) | Contact sites (protein 2) | Combination type |
|---|---|---|---|---|---|
| | | | | | Hydrogen bond |
| NOLC1 | TP53 | –236.13 | SER-60, SER-12, LYS-59, TYR-15, ASP-52, ARG-23, PRO-11, PHE-56 | GLN-104, ARG-283, ASN-131, SER-269, TYR-126, HIS-115, SER-116, LEU-289, TRP-146 | Hydrophobic interaction |

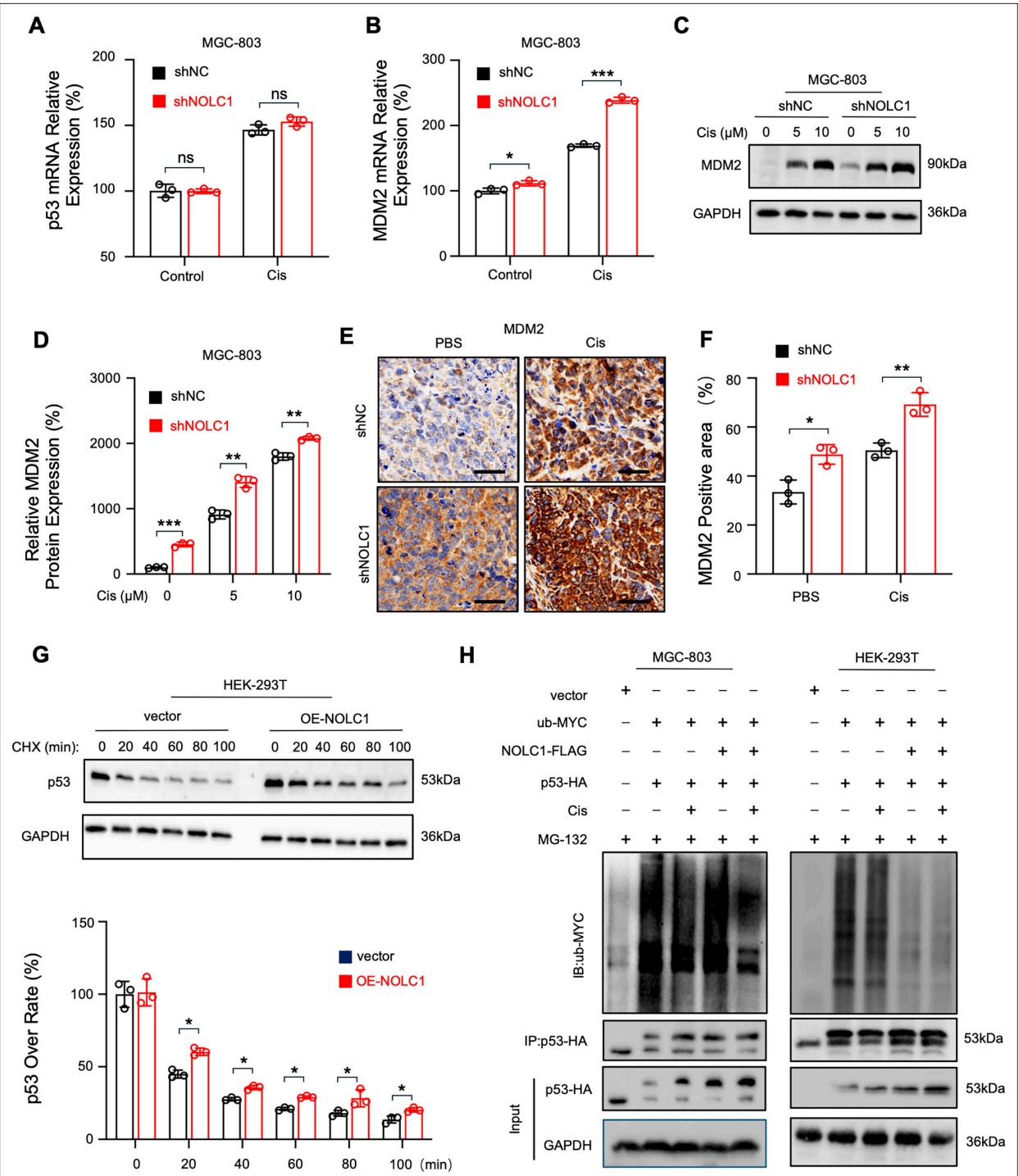

**Figure 6.** NOLC1 inhibited p53 ubiquitination and degradation. (**A**) Real-time quantitative reverse transcription polymerase chain reaction (RT-qPCR) analysis of *TP53* mRNA in MGC-803 cells transduced with sh*NC* or sh*NOLC1* lentivirus and treated with phosphate-buffered saline (PBS) or cisplatin (Cis) (15 μM) (n=3). (**B**) RT-qPCR analysis of *MDM2* mRNA in MGC-803 cells transduced with sh*NC* or sh*NOLC1* lentivirus and treated with PBS or Cis (15 μM) (n=3). (**C, D**) Western blotting (WB) assay of MDM2 in MGC-803 cells transduced with sh*NC* or sh*NOLC1* lentivirus and treated with PBS or Cis (15 μM); (**C**) Representative images and (**D**) relative MDM2 protein levels (n=3). (**E, F**) Immunohistochemical (IHC) staining of MDM2 in MGC-803 tumors from BALB/c-nu mice; (**E**) Representative images; scale bar = 50 μm. (**F**) MDM2-positive area in MGC-803 tumors (n=3). (**G**) p53 turnover rate analysis in HEK-293T cells transfected with vector or NOLC1 plasmids and treated with CHX (40 μM) for the indicated times (n=3). (**H**) Ubiquitination assay in MGC-803 and HEK-293T cells transfected with the indicated plasmids. The data are presented as the means ± SDs. ns, nonsignificant; *p<0.05; **p<0.01; ***p<0.001.

*Figure 6 continued on next page*

*Figure 6 continued*

The online version of this article includes the following source data and figure supplement(s) for figure 6:

**Source data 1.** Original western blots for *Figure 6C, G, and H*, indicating the relevant bands.

**Source data 2.** Original files for western blot displayed in *Figure 6C, G, and H*.

**Figure supplement 1.** NOLC1 promoted cisplatin (Cis) resistance by blocking p53 function in gastric cancer (GC).

**Figure supplement 1—source data 1.** Original western blots for *Figure 6—figure supplement 1D*, indicating the relevant bands.

**Figure supplement 1—source data 2.** Original files for western blot displayed in *Figure 6—figure supplement 1D*.

*MDM2* mRNA and protein levels were both increased after Cis treatment, and in the knockdown groups, the levels were greater than those in the NC groups, indicating that NOLC1 knockdown also promoted p53 transcription to *MDM2*. Similarly, the protein level of MDM2 in vivo was also elevated in the knockdown group, as detected by IHC (*Figure 6E and F*). We next examined the p53 turnover rate. In the OE-NOLC1 group, the p53 turnover rate was much slower than that in the vector group (*Figure 6G*). Furthermore, ubiquitination assays indicated that NOLC1 inhibits p53 ubiquitination and that Cis treatment induces a marked decrease in the p53 ubiquitination level in both MGC-803 and HEK-293T cells (*Figure 6H*). Collectively, these data indicate that NOLC1 inhibits p53 degradation by blocking the nuclear accumulation and transcriptional activity of p53. Therefore, the p53-MDM2 negative feedback loop is interrupted.

## p53 knockdown rescued the sensitizing effect of NOLC1 knockdown

Given that NOLC1 can promote GC resistance and interact with p53 to inhibit p53 transcription, whether NOLC1 promotes resistance by inhibiting p53 functions is still unclear. Thus, we next used siRNA to knock down the p53 protein level in MGC-803 cells. An Annexin V-APC/7-AAD apoptosis assay revealed that NOLC1 single-knockdown cells were more sensitive to Cis than were NC cells. However, the percentage of dead cells in the p53/NOLC1 double-knockdown group was not significantly different from that in the NC group (*Figure 6—figure supplement 1A and B*). Furthermore, CCK-8 assay showed similar results (*Figure 6—figure supplement 1C*). These data show that NOLC1 promotes Cis resistance via mediating p53 functions. Moreover, compared to the NOLC1 single-knockdown group, the GPX4 protein level was increased in the p53/NOLC1 double-knockdown group (*Figure 6—figure supplement 1D*). These data demonstrate that NOLC1 inhibits ferroptosis via regulating p53.

## NOLC1 knockdown increases the efficacy of anti-PD-1 combined with Cis

Considering ferroptosis can promote ICD by releasing DAMPs (*Catanzaro et al., 2024*), we further investigated whether NOLC1 knockdown can increase ferroptosis-mediated ICD. First, DAMPs, including LDH, high-mobility group box-1 (HMGB1), and calreticulin (CRT), were further analyzed. As mentioned above, the release of LDH significantly increased in the NOLC1-knockdown group after Cis treatment (*Figure 4B*, *Figure 4—figure supplement 1D*). Consistently, the fluorescence intensity of HMGB1 in the knockdown group was markedly lower than that in the NC group, especially when cells were treated with Cis (*Figure 7—figure supplement 1A and B*). These results confirmed that a large amount of HMGB1 was released into the extracellular environment after Cis treatment in the knockdown groups. Furthermore, the CRT fluorescence intensity in the knockdown groups was greater than that in the NC groups, especially treated with Cis (*Figure 7—figure supplement 1C and D*). These data showed that NOLC1 repressed ferroptosis-mediated ICD.

To further assess the role of NOLC1 in ferroptosis-mediated immunotherapy, subcutaneous mouse forestomach carcinoma (MFC) tumor-bearing 615 mice model was established for further study. Four-week-old male 615 mice were randomly divided into two groups and were subcutaneously injected with MFC cells (sh*NC* or sh*NOLC1*). When tumor volume reached about 100 mm$^3$, each group was randomly divided into four subgroups (PBS, anti-PD-1, Cis, and Cis combined with anti-PD-1) for a total of eight groups. On the 7th, 10th, and 13th days, the mice were intraperitoneally injected with drugs (Cis, anti-PD-1 antibody) (*Figure 7A*). As shown in *Figure 7B and C* and *Figure 7—figure supplement 2A*, anti-PD-1 monotherapy had no significant effect in the presence of the immunosuppressive TME. Cis treatment effectively blocked the growth of tumors. PD-1 combined with Cis had

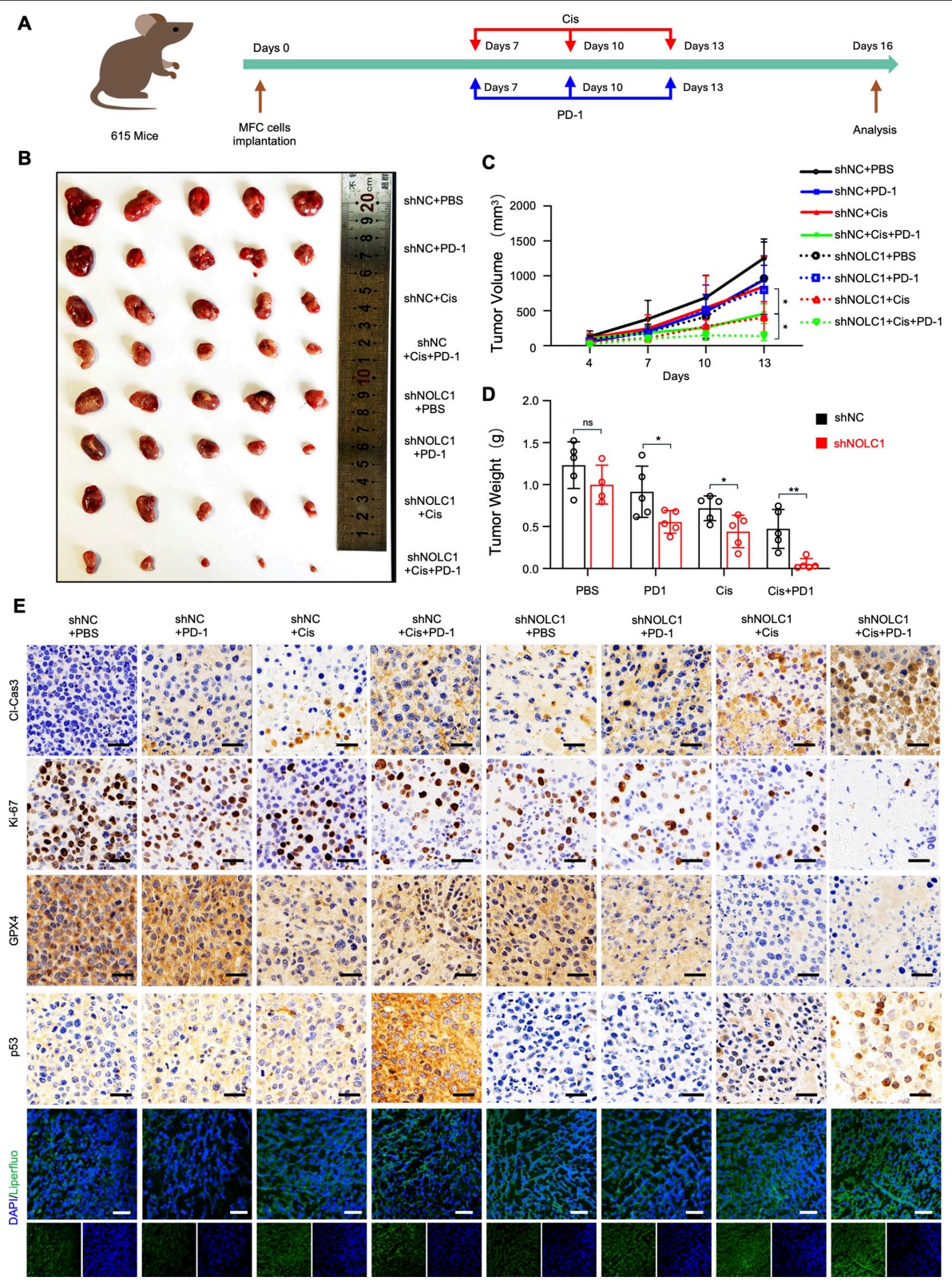

**Figure 7.** NOLC1 knockdown increased the efficacy of cisplatin (Cis) combined with anti-programmed cell death-1 (anti-PD-1) treatment. (**A**) Schematic representation showing the treatment of the subcutaneous tumors. (**B**) Corresponding tumor photographs after different treatments (n=5). (**C**) Tumor growth curves after different treatments (n=5). (**D**) Tumor weight after the indicated treatment (n=5). (**E**) Representative images of Liperfluo staining and

*Figure 7 continued on next page*

Figure 7 continued

immunohistochemical (IHC) analysis of cleaved caspase-3, Ki-67, GPX4, and p53 in tumor tissues; IHC analysis scale bar = 25 µm, Liperfluo staining scale bar = 50 µm. The data are presented as the means ± SDs. ns, nonsignificant; *p<0.05; **p<0.01.

The online version of this article includes the following figure supplement(s) for figure 7:

**Figure supplement 1.** NOLC1 knockdown increased the release of damage-associated molecular patterns (DAMPs) induced by ferroptosis.

**Figure supplement 2.** NOLC1 knockdown increased the combination treatment efficacy of anti-programmed cell death-1 (anti-PD-1) plus cisplatin (Cis).

the best efficacy. Importantly, NOLC1 silencing obviously increased the efficacy of Cis alone and anti-PD-1 plus Cis. Accordingly, the tumor weight data showed similar trends (*Figure 7D*). Subsequently, the therapeutic effect was confirmed by H&E and IHC staining. H&E staining revealed the same trend: the NOLC1-knockdown group treated with Cis plus anti-PD-1 presented the most severe necrosis in the tumor tissues (*Figure 7—figure supplement 2B and C*). As shown in *Figure 7E* and *Figure 7—figure supplement 2D–F*, Cis therapy increased the cleaved caspase-3 level and decreased the Ki-67 and GPX4 levels. The Cis plus anti-PD-1 group presented more significant changes than the Cis group did. Compared with those in the NC groups, the NOLC1-knockdown groups had higher cleaved caspase-3 and lower GPX4 and Ki-67 expressions. These findings demonstrated that NOLC1 knockdown increased the efficacy of the Cis monotherapy and combination treatment. Additionally, the knockdown group treated with anti-PD-1 plus Cis presented the highest level of nuclear p53 staining (*Figure 7—figure supplement 2G*), indicating that NOLC1 also mediates p53 nuclear accumulation in the MFC tumor model. Liperfluo is specifically oxidized by lipid peroxides and emits strong fluorescence. Liperflou staining revealed that the NOLC1-knockdown groups treated with Cis plus PD-1 presented the most intense fluorescence (*Figure 7E*). The HMGB1 level in the serum was also increased in the knockdown group treated with anti-PD-1 plus Cis (*Figure 7—figure supplement 2H*), indicating that NOLC1 suppresses ferroptosis in MFC tumors. Collectively, these data suggest that NOLC1 decreases the antitumor efficacy of anti-PD-1 plus Cis therapy via suppressing Cis-induced ferroptosis.

## Silencing NOLC1 promotes reprogramming of the TME by Cis combined with anti-PD-1 immunotherapy

The TME plays an important role in GC progression and therapeutic outcome and is correlated with the patient response to immunotherapy (*Tang et al., 2021*). In addition, ferroptosis can also reprogram the TME to increase immunotherapy efficacy (*Lei et al., 2024*). Therefore, we next explored the effect of NOLC1 knockdown on the TME. First, we collected blood samples and extracted lymphocytes. FACS was subsequently used to measure the abundance of specific lymphocytes. As shown in *Figure 8A*, the proportions of CD3⁺CD8⁺ cytotoxic T lymphocytes (CTLs), which have been reported to play pivotal roles in killing cancer cells (*Peng et al., 2024*), are significantly increased after being treated with Cis plus anti-PD-1, and the proportions of CTLs in the knockdown group are much higher than those in the NC group. In addition, Cis monotherapy could increase the proportion of CTLs in the knockdown group, but not in the NC group. Furthermore, the IF staining results also revealed an increase in the number of CTL infiltration in tumors in the knockdown group, especially in those treated with Cis plus anti-PD-1 (*Figure 8B*). The levels of tumor necrosis factor-α (TNF-α), interferon-γ (IFN-γ), and interleukin 6 (IL-6) in the combination treatment groups were significantly increased, and monotherapy did not effectively increase the levels of these factors (*Figure 8C*). In the knockdown groups, this activation was much more obvious. Taken together, these data demonstrate that NOLC1 knockdown can promote ferroptosis-mediated ICD and reprogram the TME.

## Biosafety of combined therapy with Cis and PD-1

Despite the efficacy of immunotherapy combined with chemotherapy, there are still many side effects due to the toxicity of immune checkpoint inhibitors (ICIs) or chemotherapeutic agents (*Kennedy and Salama, 2020*). To further assess the biosafety of combination therapy in vivo, we collected major organs and blood from the mice for further analysis. Analysis of liver function markers, such as alanine transaminase (ALT) and aspartate aminotransferase (AST), and kidney function markers, such as blood urea nitrogen (BUN), indicated that no significant renal or hepatic toxicity was induced by the treatment (*Figure 8—figure supplement 2A*). The organs were assessed via H&E staining. Compared with

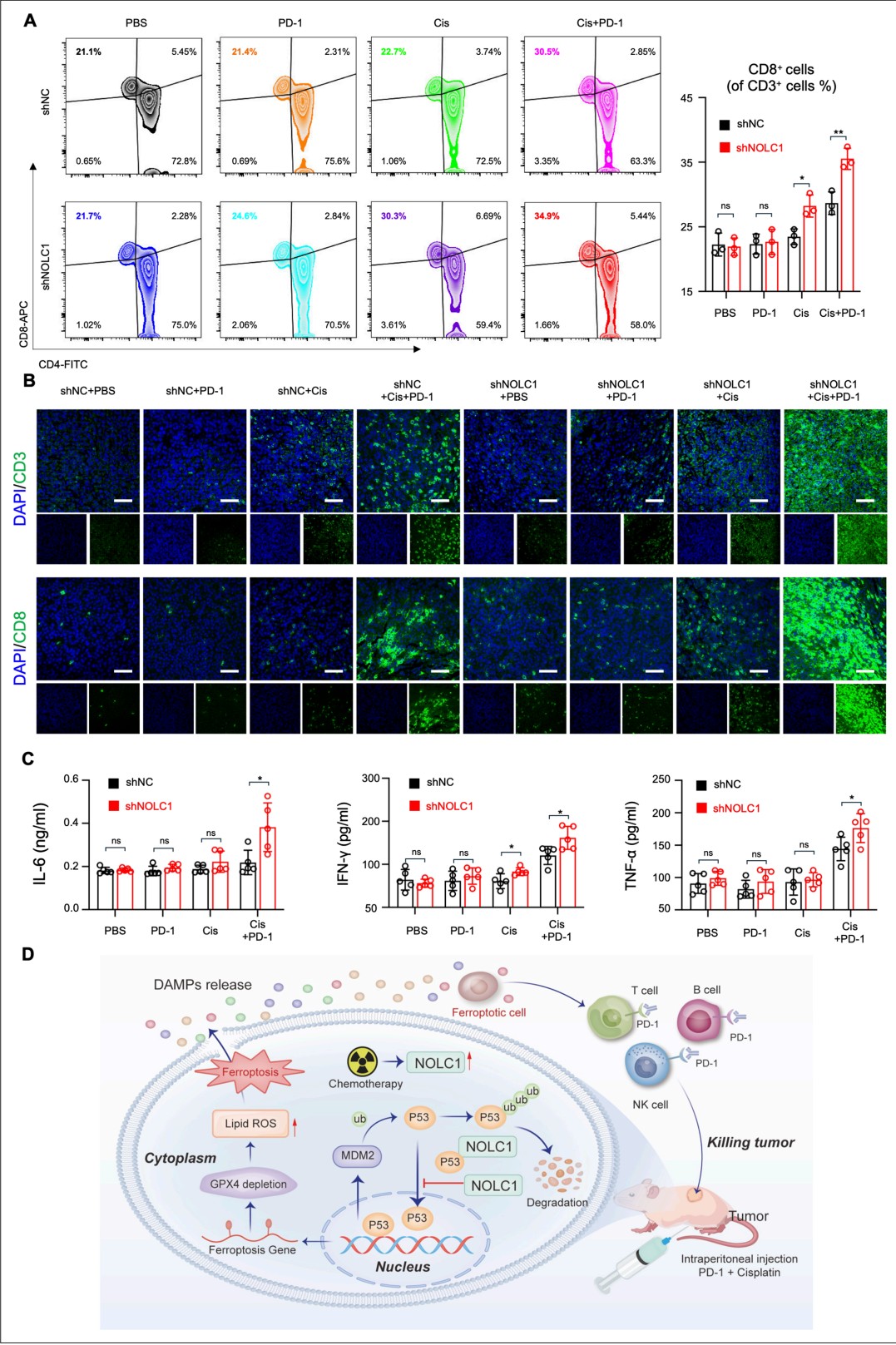

**Figure 8.** Knockdown of NOLC1 reprogrammed the tumor microenvironment after cisplatin (Cis) and anti-programmed cell death-1 (anti-PD-1) combination therapy. (**A**) Fluorescence-activated cell sorting (FACS) analysis of peripheral blood lymphocytes (n=3). (**B**) Representative immunofluorescence staining of CD-3 and CD-8 in MFC tumors after indicated treatments; scale bar = 50 μm. (**C**) The serum levels of inflammatory factors (interleukin 6 [IL-

*Figure 8 continued on next page*

*Figure 8 continued*

6], interferon-γ [IFN-γ], and tumor necrosis factor-α [TNF-α]) (n=5). (**D**) Schematic illustration of the mechanism by which NOLC1 inhibits p53-mediated ferroptosis to increase the efficacy of anti-PD-1 plus Cis in gastric cancer (GC). The data are presented as the means ± SDs. ns, nonsignificant; *p<0.05; **p<0.01.

The online version of this article includes the following figure supplement(s) for figure 8:

**Figure supplement 1.** Gating strategy of flow cytometry for peripheral blood lymphocyte analysis.

**Figure supplement 2.** Biosafety of anti-programmed cell death-1 (anti-PD-1) and cisplatin (Cis) combination therapy.

PBS treatment, combination therapy did not induce significant morphological changes or inflammation in the heart, liver, spleen, lung, or kidney (*Figure 8—figure supplement 2B*). Collectively, these data indicate the satisfactory biocompatibility of the combination therapy.

## Discussion

Despite advances in GC therapy, chemotherapy resistance remains the major obstacle to improving patient outcomes (*Smyth et al., 2020*). In this study, we identified NOLC1 as a mediator of anti-PD-1 plus Cis resistance in GC. The functions of NOLC1 in cancer are controversial, as evidenced by its pro- or antitumor potential in different tumor types and models. NOLC1 functions as an oncogene in various tumors via different pathways. In esophageal cancer (ESCA), NOLC1 can activate the PI3K-AKT signaling pathway (*Kong et al., 2021*). In ovarian cancer (OC), circ-NOLC1 binds to ESRP1 and upregulates CDK1 and RhoA expression to promote OC progression (*Chen et al., 2021b*). More importantly, NOLC1 promotes NSCLC resistance to multiple drugs. However, NOLC1 acts as a tumor suppressor in clear cell renal cell carcinoma (*Wu et al., 2021*). The results of this study indicate that NOLC1 is upregulated in GC tumors and Cis-resistant GC cells. In addition, silencing NOLC1 affected the efficacy of Cis and anti-PD-1 plus Cis combination treatment.

To determine its mechanism of action, we performed TEM imaging. The results revealed that after NOLC1 knockdown, cells displayed classical morphological features of ferroptosis. Moreover, treatment with a ferroptosis inhibitor reversed the Cis-sensitizing effect caused by NOLC1 knockdown. These findings demonstrated that NOLC1 inhibits Cis-induced ferroptosis. As a molecular chaperone, NOLC1 can interact with other proteins and regulate their translocation. We also revealed that NOLC1 interacts with p53 and inhibits p53 nuclear accumulation stimulated by Cis treatment rather than regulating p53 total protein levels, thus inhibiting p53 transcriptional activity. Eventually, p53-mediated ferroptosis is suppressed.

The mechanisms by which p53 promotes ferroptosis remain to be fully explored. Ferroptosis, a novel type of RCD, has received considerable attention in both basic and clinical research (*Zhang et al., 2022*). Recently, studies have revealed that p53 can significantly promote ferroptosis via different mechanisms. For example, p53 could transcriptionally and untranscriptionally mediate SLC7A11, a key component of the cystine/glutamate antiporter (*Shen et al., 2018*; *Wang et al., 2019*). Additionally, p53 can transcriptionally mediate several key ferroptosis proteins, such as TIGAR and GLS2 (*Jiang et al., 2015*). In this study, we found that inhibiting p53 transcriptional functions could markedly suppress ferroptosis in GC cells. These findings indicate that the transcriptional functions of p53 are critical for mediating ferroptosis.

Protein-DNA interactions are central to the response of the tumor suppressor p53 to numerous stress signals. Under various stresses, p53 can enter the nucleus and bind to specific DNA via interactions between response elements and the p53 DBD (*Senitzki et al., 2021*). Our results showed that NOLC1 interacts with the p53 DBD, which could block p53 binding to the DNA sequence, eventually inhibiting its tumor-suppressive functions. By using ChIP-seq, we found that NOLC1 strongly inhibited p53 binding to the transcription-associated DNA sequence (*Figure 5M*). Our results show that the interaction between NOLC1 and p53 not only inhibits p53 binding to the DNA sequence but also decreases p53 nuclear accumulation. As shown in *Figure 5—figure supplement 1A and B*, the interaction between NOLC1 and p53 was limited to the cytoplasm rather than the nucleus. Thus, NOLC1-mediated inhibition of p53 functions relies mainly on blocking p53 nuclear accumulation rather than inhibiting its interaction with DNA.

Traditionally, the ability of tumors to inhibit p53 functions relies primarily on the MDM2-p53 negative feedback loop or mutant p53 functions (*Chen et al., 2020*). Our results reveal a novel way to inhibit p53 functions: sequestration of the p53 protein in the cytoplasm. p53 is mutated in 50% of cancers, and most of these mutations occur in the DBD (86% of tumorigenic mutants are present in the DBD) (*Leroy et al., 2013*; *Strano et al., 2007*). Therefore, this suppressive effect of NOLC1 on p53 might be limited to only WT p53, and this combination of p53 and NOLC1 could further inhibit p53 functions.

More and more studies have shed light on the important role of p53 in immunotherapy. The activation of p53 suppresses tumor immune evasion and promotes the antitumor immune response via the cGAS-STING pathway, regulating T-cell-mediated antitumor immunity, etc. (*Concepcion et al., 2022*). In addition, mut p53 exerts immunosuppressive effects via multiple mechanisms, such as increasing tumor-associated neutrophil infiltration, decreasing CTL infiltration, and promoting fibrosis (*Dong et al., 2024*). Our results demonstrated that activating p53 transcriptional functions could enhance the immune response and reprogram the TME by promoting ferroptosis. Current treatments for GC have evolved significantly through systemic therapies and include chemotherapy, targeted therapy, and immunotherapy. Immunotherapy has substantially advanced the treatment of GC with microsatellite instability (MSI) but has limited efficacy for GC with microsatellite stability (MSS) (*Guan et al., 2023*). However, MSI status is observed for only approximately 20% of GC patients (*Cristescu et al., 2015*; *Puliga et al., 2021*). Thus, identifying new molecular biomarkers is highly important. In this study, we found that NOLC1 knockdown could transform the chemoresistance tumor environment to a sensitive state. Thus, lower expression of NOLC1 could be a marker for identifying patients with a better response to the combination treatment of anti-PD-1 plus Cis. Additionally, our studies identified NOLC1 could be a promising target for increasing ICI sensitivity in MSS GC.

In conclusion, our study demonstrated that NOLC1 binds to the p53 DBD and inhibits p53 nuclear accumulation and transcriptional functions, thereby inhibiting ferroptosis and ferroptosis-mediated ICD. Low expression of NOLC1 could be a new biomarker for identifying GC patients who may benefit from Cis plus anti-PD-1 treatment, and targeting NOLC1 could increase the treatment response of anti-PD-1 plus Cis.

## Materials and methods

### Expression plasmids

Expression plasmids encoding Flag, HA- or MYC-tagged human NOLC1, p53, and Ub, in addition to reporter vectors containing the p53 promoter, were purchased from RiboBio. Lentivirus-NOLC1 was obtained from Shanghai Genechem Co.,Ltd. All coding sequences were verified by DNA sequencing.

### Chemicals

Ferrostatin-1, Cis, MG-132, and CHX were obtained from MCE.

### Antibodies

NOLC1 (Cat: DF4264, RRID:AB_2836615), PARP (Cat: DF7198, RRID:AB_2839150), cleaved-PARP (Asp214) (Cat: AF7023, RRID:AB_2835327), P21 (Cat: AF6290, RRID:AB_2827699), FAS (Cat: AF5342, RRID:AB_2837827), MDM2 (Cat: AF0208, RRID:AB_2833395), and ACSL4 (Cat: DF12141, RRID:AB_2844946) antibodies were obtained from Affinity. Lamin B1 antibody (Cat. 12987-1-AP, RRID:AB_2136290), MYC tag antibody (Cat. 16286-1-AP, RRID:AB_11182162), p53 antibody (Cat. 60283-2-Ig, RRID:AB_2881401), caspase-3/p17/p19 antibody (Cat. 19677-1-AP, RRID:AB_10733244), GAPDH antibody (Cat. 10494-1-AP, RRID:AB_2263076), PTEN antibody (Cat. 22034-1-AP, RRID:AB_2878977), DYKDDDDK tag antibody (Cat. 66008-4-Ig, RRID:AB_2918475), and HA-tag antibody (Cat. 51064-2-AP, RRID:AB_11042321) were obtained from Proteintech. Bcl-2 antibody (Cat: ab1828585), CD3 epsilon antibody (Cat: ab16669, RRID:AB_443425), and CD8 alpha antibody (Cat: ab217344, RRID:AB_2890649) were obtained from Abcam. Phospho-histone H2A.X (Ser139) antibody (Cat: 2577, RRID:AB_2118010), GPX4 antibody (Cat: 52455, RRID:AB_2924984), Ki-67 antibody (Cat: 9129, RRID:AB_2687446), and FSP1 antibody (Cat: 24972, RRID:AB_3090192) were obtained from CST. The donkey anti-mouse IgG (H+L) highly cross-adsorbed secondary antibody Alexa Fluor 647 (Cat: A31571, RRID:AB_162542), goat anti-rabbit IgG (H+L) cross-adsorbed secondary antibody

Alexa Fluor 555 (Cat: A21428, RRID:AB_141784), goat anti-rabbit IgG (H+L) secondary antibody, HRP (Cat: 31460, RRID:AB_228341), and goat anti-mouse IgG (H+L) secondary antibody HRP (Cat: 31430, RRID:AB_228307) were obtained from Invitrogen. The InVivoMAb anti-mouse PD-1 antibody (Cat: BE0146, RRID:AB_10949053) was obtained from Bio X Cell.

## Kits

Cell Counting Kit-8 (CCK-8) and a Cytotoxicity LDH Assay Kit (Cat: CK12) were obtained from Dojindo Laboratories. A dual-luciferase kit and $H_2O_2$ assay kit were obtained from Promega. Enzyme-linked immunosorbent assay (ELISA) kits for HMGB-1, IL-6, TNF-α, IFN-γ, ALT, AST, and BUN were obtained from Jiangsu Meimian Industrial Co., Ltd. mRNA reverse transcription kits were obtained from Takara.

## Probes

The DCFH-DA probe was obtained from Thermo Fisher Scientific. JC-1 probes were obtained from Beyotime Biotechnology. MitoPeDPP and Liperfluo probes were obtained from Dojindo Laboratories.

## Primes

| NOLC1-F | TTCCTGCGCGATAACCAACTC |
|---|---|
| NOLC1-R | CCTGTAACTTTCGCTCTGGGA |
| CDKN1A-F | TGTCCGTCAGAACCCATGC |
| CDKN1A-R | AAAGTCGAAGTTCCATCGCTC |
| BAX-F | CCCGAGAGGTCTTTTTCCGAG |
| BAX-R | CCAGCCCATGATGGTTCTGAT |
| PTEN-F | AGGGACGAACTGGTGTAATGA |
| PTEN-R | CTGGTCCTTACTTCCCCATAGAA |
| FAS-F | AGATTGTGTGATGAAGGACATGG |
| FAS-R | TGTTGCTGGTGAGTGTGCATT |

## siRNA

| si-NC | UUCUCCGAACGUGUCACGU |
|---|---|
| si-h-NOLC1-1 | CAAGAAGACUGUACCUAAA |
| si-h-NOLC1-2 | CCAAGAAUUCUUCAAAUAA |
| si-h-NOLC1-3 | CAUCUAAGUCUGCAGUUAA |
| si-h-P53-1 | GCAUCUUAUCCGAGUGGAAGGTT |
| si-h-p53-2 | ACUACAACUACAUGUGUAACATT |
| si-h-p53-3 | AGCGAGCACUGCCCAACAACATT |

## Cis-resistant cell line construction

MGC-803 cells were treated with Cis (1 µM) for 3 months, followed by treatment with Cis (5 µM) for 3 months.

## Cell transfection and lentiviral infection

MKN-45 and MGC-803 cells with stable NOLC1 expression at lower levels were generated by transducing a lentiviral vector followed by the ORF of NOLC1. Stable cells were selected using puromycin (2 µg/mL) for 48 hr.

## Cell culture

HEK293T, MGC-803, and MKN-45 cell lines were purchased from the National Infrastructure of Cell Line Resource (Beijing, China) and authenticated via STR profiling. All cells were cultured in DMEM

(Gibco, USA) supplemented with 10% fetal bovine serum (FBS; Gibco, USA) and 1% anti-anti (100 U/mL; Gibco, USA). For Cis-resistant cell line construction, MGC-803 cells were treated with Cis (1 µM) for 5 months, followed by treatment with Cis (5 µM) for 1 month. All the cells were maintained at 37°C in a 5% $CO_2$ cell culture incubator.

## RNA isolation and RT-qPCR

According to the manufacturer's instructions, RNA was extracted from cells and tissues using TRIzol (Invitrogen, USA). Subsequently, RNA was reverse-transcribed into cDNA using a reverse transcription kit (Takara, Dalian, China). The expression levels of RNA transcripts were analyzed using a Bio-Rad CFX96 Real-Time PCR System (Bio-Rad, USA). All samples were normalized to GAPDH.

## Western blot

Cell samples were collected and washed twice with cold PBS. After centrifugation for 5 min, cells were lysed with RIPA lysis buffer (Beyotime, China) containing 1% PMSF and incubated on ice for 40 min. After centrifugation at 13,000 rpm for 20 min at 4°C, the protein supernatant is collected. Proteins were fractionated by SDS-PAGE, transferred to PVDF membranes, blocked in 5–10% nonfat milk in TBS-Tween-20, and then treated with specific primary and secondary antibodies for 12 hr at 4°C. Finally, the blots were detected using enhanced chemiluminescence detection (NCM China). Intensities were analyzed using ImageJ software, and relative expression was normalized to that of GAPDH.

## H&E and IHC staining assay

The tumor and organ samples were collected and fixed in 4% paraformaldehyde solution for 24 hr at 4°C. Then, dehydrated, paraffin-embedded, sectioned, and stained with H&E.

For IHC staining analysis, sections were repaired in sodium citrate buffer and subsequently blocked with $H_2O_2$ for 30 min. Then, sections were blocked with 5% normal goat serum, 0.1% Triton X-100, and 3% $H_2O_2$ for 30 min at room temperature and subsequently incubated with the appropriate primary antibodies overnight at 4°C. Finally, IHC staining was performed with horseradish peroxidase (HRP) conjugates using DAB detection.

## CCK-8 assay

MGC-803 or MKN-45 cells were seeded in 96-well plates and incubated overnight. The cells were then treated with the indicated drugs for 24 hr. Afterward, 10% CCK-8 (Dojindo, Japan) was added, and the cells were incubated at 37°C for 1–3 hr. The absorbance of each well was measured by a microplate reader (Tecan Switzerland) set at 450 nm. All experiments were performed in triplicate.

## Colony formation assay

For the colony formation assay, $1 \times 10^3$ MGC-803 cells or $2 \times 10^3$ MKN-45 cells are seeded into each well of six-well plates. The medium was renewed every 3 days. The colonies were fixed with methanol after 2 weeks and then stained with 0.1% crystal violet (Sigma-Aldrich, USA) for 30 min and washed twice with PBS. The number of colonies with more than 50 cells was counted.

## Cell apoptosis assay

MGC-803 or MKN-45 cells were plated in six-well plates and incubated overnight. The cells were then treated with the indicated treatment. Cell apoptosis was detected using an Annexin V-APC apoptosis detection kit (Multi Science, China, Hangzhou) according to the manufacturer's protocols. Briefly, the collected cells were washed twice with PBS and with binding buffer. Cells were resuspended in 500 µL binding buffer containing 5 µL of Annexin V-APC and 10 µL of 7-AAD. The analysis was then carried out using flow cytometry (Beckman Coulter, USA).

## LDH release assay

MGC-803 or MKN-45 cells were plated in 96-well plates ($1 \times 10^4$ cells/well) and incubated overnight. The cells were then treated with the indicated treatment. Extracellular LDH was detected using an LDH assay kit (Dojindo, Japan) according to the manufacturer's protocols.

## IF analysis

MGC-803, MKN-45, or HEK-293T cells were seeded in glass-bottom cell culture dishes ($2\times10^5$ cells/well) and incubated for 12 hr. Cells were then treated with PBS or Cis for 24 hr. Cells were then fixed with 4% paraformaldehyde for 20 min and blocked with 3% bovine serum albumin in PBS for 1 hr. The fixed cells were then incubated with primary antibodies (NOLC1, p53, anti-HA tag, anti-Flag tag, HMGB1, γ-H2AX, and CRT) at 4°C for 12 hr. Subsequent incubation with an Alexa Fluor 562-conjugated anti-mouse or Alexa Fluor 647-conjugated anti-rabbit secondary antibody at room temperature for 2 hr. DAPI was used for nuclear staining. IF images were captured using a confocal microscope (Nikon Japan) and analyzed using ImageJ software.

## FCM analysis of total ROS, MitoPeDPP, and mitochondrial membrane potential

MGC-803 and MKN-45 cells were treated with PBS or the indicated Cis concentration for 48 hr. The cells were then harvested and resuspended in 500 μL of PBS containing 1 μM DCFH-DA (Thermo Fisher, USA), 5 μM MitoPeDPP (Dojindo, Japan), or 1 μM JC-1 (Beyotime, China) and incubated at 37°C for 30 min. The cells were analyzed by flow cytometer (Beckman Coulter, USA).

## Intracellular total ROS, MitoPeDPP, and mitochondrial membrane potential fluorescence imaging

Intracellular ROS were detected using a DCFH-DA probe (Thermo Fisher, USA). Cells were harvested and treated with DCFH-DA (1 μM) at 37°C for 40 min. Fluorescence signals were detected using a confocal microscope (Nikon Japan). Intracellular mito-ROS were detected using the MitoPeDPP probe. The harvested cells were isolated and treated with MitoPeDPP (5 μM) for 1 hr at 37°C. The fluorescence signals were detected using a confocal microscope (Nikon Japan). Mitochondrial membrane potential was detected with a JC-1 probe (Beyotime, China). The collected cells were treated with JC-1 (1 μM) for 20 min at 37°C. The fluorescence signal was then detected using a confocal microscope (Nikon, Japan).

## $H_2O_2$ content assay

Total intracellular $H_2O_2$ was detected using a ROS-Glo $H_2O_2$ Assay Kit (Promega, USA). Briefly, MGC-803 or MKN-45 cells were plated at the desired density in ≤80 μL of medium in 96-well test plates with opaque walls. Cells were treated with the indicated concentrations of Cis for 24 hr. Then, 20 μl $H_2O_2$ substrate solution was added to the cell culture medium and incubated for 6 hr. ROS-Glo Detection Solution was then added and incubated for 20 min.

## Transmission electron microscopy

MGC-803 cells were seeded in six-well plates and treated with PBS or Cis for 24 hr. Then, the cells were collected and fixed with 3% glutaraldehyde in 0.1 M phosphate buffer, then fixed with $OsO_4$. The cells were then dehydrated and sectioned at 60–80 nm. Subsequently, the cells were stained with uranyl acetate and lead nitrate. Finally, the sections were observed using a JEM-1230 transmission electron microscope (JEOL, Japan).

## Coimmunoprecipitation

HEK-293T cells were seeded in 6 cm cell culture dishes. When they reached 70% density, the cells were transfected with the p53-HA, NOLC1-Flag, or Ub-MYC plasmid. Cells were collected 48 hr later and lysed using IP lysis buffer (containing 1% protease inhibitor) for 30 min on ice. Then, anti-Flag magnetic beads were used to immunoprecipitate the NOLC1-Flag protein. The protein-antibody-magnetic bead complexes were dissociated by boiling using a metal bath for 10 min. The absorbed proteins were resolved by SDS-PAGE and subjected to immunoblotting with the indicated primary and secondary antibodies. The cell lysates were analyzed as an input.

## Chromatin immunoprecipitation

HEK293T cells were seeded in 6 cm cell culture dishes. When they reached 70% density, the cells were transfected with p53-HA or NOLC1-Flag plasmids. The cells were harvested and processed

with a SimpleChIP Enzymatic Chromatin IP Kit (CST, USA) following the manufacturer's protocol. Briefly, after being fixed with 1% formaldehyde for 10 min at room temperature and lysed with Lysis Buffer-Protease Inhibitor solution, the cells were digested with micrococcal nuclease and immuno-precipitated with rabbit anti-HA antibodies. Then, DNA-protein complexes were eluted in extraction buffer and incubated overnight at 65°C to reverse the cross-links. Finally, DNA was purified using spin columns for sequencing. DNA libraries and sequences were performed by Cosmos Wisdom Biotech Co., Ltd (Hangzhou, China).

## mRNA sequencing
MGC-803 and MGC-803-CR cells were seeded in 6 cm cell culture dishes. When reached 70% density, extracted by TRIzol (Thermo Fisher, USA). RNA integrity was evaluated with a 1.0% agarose gel. Then, RNA samples were sent to Cosmos Wisdom Biotech Co., Ltd (Hangzhou, China) for sequencing.

## Luciferase reporter assay
HEK293T cells were seeded in 6 cm cell culture dishes. When they reached 70% density, the cells were transfected with indicated reporters bearing an ORF encoding firefly luciferase, and pRL-Luc containing the Renilla luciferase ORF as the internal control for transfection. Briefly, luciferase assays were performed using a dual-luciferase assay kit (Promega), luciferase activity was quantified with a microplate reader, and firefly luciferase activity was normalized to Renilla luciferase activity as the internal control.

## Animal experiments
The animal experiment procedures were approved by the Institutional Animal Care and Use Committee of Wenzhou Institute, University of Chinese Academy of Science (approval number: WIUCAS23112802). The animal research data was collected and reported according to ARRIVE guidelines. 4-week-old male BALB/c-nu mice were purchased from Vital River Laboratory Animal Technology Co., Ltd. (Shanghai). 4-week-old male 615 mice were purchased from ZiYuan Laboratory Animal Technology Co., Ltd. (Hangzhou). All mice were raised in specific pathogen-free animal rooms.

To establish the MGC-803 tumor-bearing model, an MGC-803 tumor-bearing model was established by subcutaneous injection of lentivirus (shNC or shNOLC1)-transfected MGC-803 cells ($5 \times 10^6$) into the right flanks of BALB/c male athymic nude mice. When the tumor volume reached about 80 mm$^3$, the mice were randomly divided into two groups and intravenously administered PBS or Cis (40 μg/mouse) every 3 days for a total of six times.

To establish the MFC tumor-bearing model, an MFC tumor-bearing model was established by subcutaneous injection of lentivirus (shNC or shNOLC1)-transfected MFC cells ($2 \times 10^6$) into the right flanks of 615 male mice. When the tumor volume reached about 100 mm$^3$, the mice were randomly divided into four groups (n=5) and intravenously administered PBS, PD-1 (100 μg/mouse), Cis (40 μg/mouse), or PD-1 combined with Cis every 3 days for a total of three times.

## Enzyme-linked immunosorbent assay
Blood samples were collected and solidified at 4°C for 2 hr. Then, the samples were centrifuged at 2000 × $g$ for 20 min at 4°C to obtain the blood serum. Cytokine and blood biochemistry levels were detected using ELISA kits according to the manufacturer's protocol.

## In vivo Liperfluo analysis
Freshly frozen tissue was sectioned at a thickness of 5 μm, and then the slices were incubated with a Liperfluo fluorescent probe for 1 hr at 37°C, followed by nuclear staining with Hoechst 33342. Finally, the slices were observed using confocal microscopy.

## Peripheral blood lymphocyte analysis
To analyze the peripheral blood lymphocytes, blood samples were collected to obtain lymphocytes. Blood was separated and filtered to obtain a single lymphocyte suspension. After that, the lymphocytes were blocked with Mice TruStain FcX (Fc Receptor Blocking Solution, BioLegend) for 10 min. Then, cells were stained with a LIVE/DEAD Fixable Violet Kit (Invitrogen) and subsequently stained with the following anti-mouse antibodies: CD3-PE, CD4-FITC, CD8-APC/Cyanine7 according to the

manufacturer's methods. Finally, cells were analyzed via a flow cytometer (Beckman Coulter, USA), and data analysis was conducted using FlowJo software (BD, USA).

## Molecular docking

The crystal structure of P53 (PDB code: 8E7B) was downloaded from the RCSB Data Bank (http://www.pdb.org). The NOLC1 protein structure was constructed by HDOCK based on its amino acid sequence. Using HDOCK software, each protein was set to be rigid, the docking contact sites were set to the full surface, and the resulting conformations were set to 100 after docking. The docking score was calculated based on the expert iterative scoring function ITScorePP. A more negative docking score implies a more likely binding model. In this study, the most negative energy conformation was selected by the evaluation function and optimized by the minimization module in the MOE 2019.1 software platform to solve the unreasonable contacts in the spatial structure that may occur in rigid docking. Amber10: ETH was selected as the force field for energy minimization, and water molecules were selected for the solvation model. The optimization method was divided into two steps: steepest descent and conjugate gradient, and the maximum number of iterations was 5000. PyMOL2.1 software was used to visualize and analyze the model.

## Statistics

All data are shown as mean ± standard deviation (SD), and statistical analyses were conducted using GraphPad Prism 9 (GraphPad Software, San Diego, CA, USA). Student's t test was used to determine the significance of differences between two groups. The difference of $*p<0.05$ was considered to indicate statistical significance, and $**p<0.01$ and $***p<0.001$ indicate high significance.

## Acknowledgements

We would like to thank all the participants involved in this study. We acknowledge the efforts of Dr. Cifeng Cai of Cosmos Wisdom Biotech., Ltd. (Hangzhou, China) for sequencing services. This work was supported by grants from the National Natural Science Foundation of China (Grant NO:81972261, 82272172) and the National Key R&D Program of China (Grant NO:2023YFC2413400).

## Additional information

### Funding

| Funder | Grant reference number | Author |
|---|---|---|
| National Natural Science Foundation of China | 82272172 | Weijian Sun |
| Medical Science and Technology Project of Zhejiang Province | WKJ-ZJ-2322 | Weijian Sun |
| Medical Science and Technology Project of Zhejiang Province | KLZ25H300001 | Weijian Sun |
| National Natural Science Foundation of China | 82404783 | Xufeng Lu |

The funders had no role in study design, data collection and interpretation, or the decision to submit the work for publication.

### Author contributions

Shengsheng Zhao, Conceptualization, Data curation, Formal analysis, Supervision, Validation, Investigation, Writing – original draft, Writing – review and editing; Ji Lin, Funding acquisition, Validation, Investigation; Bingzi Zhu, Data curation, Investigation, Writing – review and editing; Yin Jin, Supervision, Investigation; Qiantong Dong, Methodology; Xiaojiao Ruan, Yongdong Yi, Jianhua Lu, Funding acquisition; Dan Jin, Danna Liang, Investigation; Binglong Bai, Formal analysis; Hongzheng Li, Data curation; Letian Meng, Xiang Wang, Yuekai Cui, Software; Yuyang Gu, Visualization; Xian

Shen, Resources, Supervision; Xufeng Lu, Resources, Supervision, Validation, Investigation, Writing – review and editing; Shangrui Rao, Conceptualization; Weijian Sun, Conceptualization, Resources, Supervision, Funding acquisition, Project administration, Writing – review and editing

### Author ORCIDs
Shengsheng Zhao (ID) https://orcid.org/0009-0001-7242-2662
Xufeng Lu (ID) https://orcid.org/0000-0002-4247-4955
Weijian Sun (ID) https://orcid.org/0000-0001-5789-1287

### Ethics
Human subjects: Studies involving human participants were viewed and approved by the Ethics Committee in Clinical Research (ECCR) of the First Affiliated Hospital of Wenzhou medical University Acceptance Number: KY2022-202.

The animal experiments procedures were approved by the Institutional Animal Care and Use Committee of Wenzhou Institute, University of Chinese Academy of Science (approval number: WIUCAS23112802).

Reviewer #1 (Public review): https://doi.org/10.7554/eLife.103904.3.sa1
Reviewer #2 (Public review): https://doi.org/10.7554/eLife.103904.3.sa2
Author response https://doi.org/10.7554/eLife.103904.3.sa3

## Additional files

### Supplementary files
MDAR checklist

Source data 1. ChIP results in HKE293T cells tranduced with p53-HA plasmid.

Source data 2. ChIP results in HKE293T cells tranduced with p53-HA and NOLC1-Flag plasmid.

### Data availability
All data generated or analyzed during this study are in the deposited in manuscript and supporting files. Source data files contain the numerical data used to generate the figures.

The following previously published dataset was used:

| Author(s) | Year | Dataset title | Dataset URL | Database and Identifier |
| --- | --- | --- | --- | --- |
| Tang Z, Li C, Kang B, Gao G, Li C, Zhang Z | 2017 | GEPIA: a web server for cancer and normal gene expression profiling and interactive analyses | http://gepia.cancer-pku.cn/detail.php?gene=NOLC1 | GEPIA, ENSG00000166197.16 |

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
