## [Editor Report · eLife Assessment]

This **fundamental** study identified a novel role of NOLC1 in regulating p53 nuclear transcriptional activity and p53-mediated ferroptosis in gastric cancer. After major revisions, the evidence supporting the conclusions is **solid**. However, some new experiments are needed to draw more robust conclusions regarding the ferroptosis-associated studies.

---

## [Referee Report · Reviewer #1 (Public review)]

Summary:

In this manuscript, the authors addressed the previous comments from reviewers.

Strengths:

This study identified that NOLC1 could bind to p53 and decrease its nuclear transcriptional activity, then inhibit p53-mediated ferroptosis in gastric cancer.

Weaknesses:

There are a few Western blot images that were processed with excessive contrast adjustment, such as Figure 2I (Caspase-3 in MKN-45 group), Figure 4H (GPX4 in MKN-45 group), and Figure 5G/5I.

---

## [Referee Report · Reviewer #2 (Public review)]

Summary:

Shengsheng Zhao et al. investigated the role of nucleolar and coiled-body phosphoprotein 1 (NOLC1) in relegating gastric cancer (GC) development and cisplatin-induced drug resistance in GC. They found a significant correlation between high NOLC1 expression and the poor prognosis of GC. Meanwhile, upregulation of NOLC1 was associated with cis-resistant GC. Experimentally, the authors demonstrate that knocking down NOLC1 increased GC sensitivity to Cis possibly by regulating ferroptosis. Mechanistically, they found NOLC1 suppressed ferroptosis by blocking the translocation of P53 from the cytoplasm to the nucleus and promoting its degradation. In addition, the authors also evaluated the effect of combinational treatment of anti-PD-1 and cisplatin in NOLC1 -knockdown tumor cells, revealing a potential role of NOLC1 in the targeted therapy for GC.

Strengths:

Chemoresistance is considered a major reason causing failure of tumor treatment and death of cancer patients. This paper explored the role of NOLC1 in the regulation of Cis-mediated resistance, which involves a regulated cell death named ferroptosis. These findings provide more evidence highlighting the study of regulated cell death to overcome drug resistance in cancer treatment, which could give us more potential strategies or targets for combating cancer.

Weaknesses:

More evidence supporting the regulation of ferroptosis induced by Cisplatin by NOLC1 should be added. Particularly, the role of ferroptosis in the cisplatin-resistance should be verified and whether NOLC1 regulates ferroptosis induced by additional FINs should be explored. Besides, the experiments to verify the regulation of ferroptosis sensitivity by NOLC1 are sort of superficial. The role of MDM2/p53 in ferroptosis or cisplatin resistance mediated by NOLC1 should be further studied by genetic manipulation of p53, which is the key evidence to confirm its contribution to NOLC1 regulation of GC and relative cell death.

---

## [Author Response]

The following is the authors’ response to the original reviews

**Reviewer #1:**
Summary:In this manuscript (eLife-RP-RA-2024-103904), the authors identified that NOLC1 was upregulated in gastric cancer samples, which promoted cancer progression and cisplatin resistance. They further found that NOLC1 could bind to p53 and decrease its nuclear transcriptional activity, then inhibit p53-mediated ferroptosis. There are several major concerns regarding the conclusions.Strengths:This study identified that NOLC1 could bind to p53 and decrease its nuclear transcriptional activity, then inhibit p53-mediated ferroptosis in gastric cancer.Weaknesses:The major conclusions were not sufficiently supported by the results. The experiments were not conducted in a comprehensive manner.Major concerns(1) The authors investigated NOLC1 expression in gastric cancer (GC) using clinical samples, which is valuable; however, the sample array includes only 3 patients. This sample size is insufficient to support conclusions for human samples. Please increase the sample size and apply a more robust statistical analysis. Additionally, specify the statistical methods used in the figure legend.

Thanks very much for the kind comments and great suggestions. As suggested, we have increased the sample size of GC patients, and the new data (six pair samples) was shown in Fig. S1A, further reflecting that NOLC1 was upregulate in gastric cancer (GC). Moreover, the statistical methods have been added in each figure legend.

(2) These data are not sufficient to support the key conclusion of this study "NOLC1 is significantly upregulated in GC tissues and Cis-resistant GC cells". There is no convincing data showing that NOLC1 upregulation is specific to cancer cells or any other cell types. Based on the following results that NOLC1 expressed in cancer cells can support cancer cell survival and drug resistance, the authors switched to investigating the role of NOLC1 in cancer cells without demonstrating cancer cells indeed highly upregulate NOLC1.

Thanks for raising this good question. As shown in Fig. 1E-F, the TCGA database have shown that NOLC1 was upregulated in GC. Moreover, we further analyzed the NOLC1 expression level in other cancer type, according to the Human Protein Atlas (https://www.proteinatlas.org/). The results indicated that NOLC1 mRNA level was much higher in almost all cancers except acute myeloid leukemia (LAML). In addition, according to the gene expression profiling interactive analysis (GEPIA, http://gepia.cancer-pku.cn/index.html), NOLC1 mRNA level was above 100 nTPM in most gastric cancer cell lines, however in most non-cancerous cell lines was below 100 nTPM, indicating that NOLC1 was up-regulated in gastric cancer.

The mRNA level of NOLC1 in different GC cells and non-cancerous cells.

(3) The authors primarily use MGC-803 cells for experiments; however, MGC-803 is known to be a HeLa-contaminated cell line. Could the authors explain this choice of using this cell line only? Did they validate key findings with additional cell lines? This is particularly important for assays such as cisplatin resistance validation, in vivo experiments, TEM imaging, and MitoPeDPP fluorescence imaging.

Thanks for raising this good question. We are not only use MGC-803 cells, the key findings in vitro was also validated in MKN-45 cells (Fig. 2), and in vivo experiment also validated in Mouse Forestomach Carcinoma cells (MFC)-tumor bearing 615 mice model (Fig 7). Furthermore, we further added some experiments in MKN-45 cells. The TEM imaging showed that NOLC1 could significantly inhibit cisplatin (Cis) induced lipid membrane damage in MKN-45 cells (Fig. S6A). Moreover, MitoPeDPP fluorescence assay analyzed by FCAs also indicating that rapid ROS was enriched in mitochondria in MKN-45 cells (Fig. 4E, Fig. S6J).

(4) In Figure 2, did the authors perform assays with NOLC1 overexpression? If so, please include these results to strengthen the conclusions.

Thanks very much for the kind comments and great suggestions. As suggested, we added new data about NOLC1 overexpression assay Cell counting kit-8 assay shows that NOLC1-overexpression group is more resistance to Cis compared to vector group (Fig. S4E, S5A).

(5) The authors show in Figures 2A-B that shNOLC1 without cisplatin treatment does not affect cell viability. However, Figures 2D-E suggest increased apoptosis in shNOLC1 cells without cisplatin treatment. Additionally, in vivo studies in Figure 3 show no significant difference between the shNC+PBS and shNOLC1+PBS groups, which appears contradictory to the apoptosis assays. Similarly, Ki67 staining shows decreased scores in the shNOLC1 group compared to shNC. Could the authors clarify this inconsistency?

Thanks for raising this good question. In Fig 2D-E, the difference in proportion of death cells between shNOLC1 and shNC treated with PBS groups were only 3% (MGC-803) and 7% (MKN-45) which is much lower than that treated with cisplatin in vitro. Moreover, in vivo analysis indicated that the average tumor volume in NOLC1+PBS group was smaller than that in NC group, but there was no statistical significance (p value = 0.3962). Moreover, tumor proliferation is a complex process regulated by many factors [1,2], thus the level of Ki67 is by no means the same as the rate of tumor proliferation, might be positively correlated.

(6) In Figure 4, NOLC1 knockdown appears to enhance cisplatin-induced ferroptosis rather than apoptosis. Given p53's role in apoptosis, did the authors compare the effects of NOLC1 on cisplatin-induced apoptosis vs. ferroptosis? If so, please clarify whether NOLC1 predominantly regulates apoptosis or ferroptosis.

Thanks for raising this good question. We do have compared the effects of NOLC1 on cisplatin-induced apoptosis vs. ferroptosis. As shown in Fig. 5A, NOLC1 knockdown obviously increased the BCL-2 protein level which is an anti-apoptotic protein and mediated by p53 via protein interaction in cytoplasm[3,4], this phenomenon may cause by the increasing level of p53 in cytoplasm (Fig. 6I). Also, the TEM imaging showed the classic ferroptotic morphological changes rather than apoptosis (Fig. 5A, S6A). Taken together, NOLC1 mainly regulates p53 mediated ferroptosis rather than apoptosis.

(7) Did the authors perform co-IP assays with p53 or HA antibodies to immunocapture NOLC1? If not, please add this experiment to support protein interactions. The mechanistic correlation between p53 and NOLC1 can be supported by adding experiments using multiple GC cell lines with various p53 alterations (such as loss-of- function or gain-of-function mutations/deletions). This is critical because the authors specifically claimed that NOLC1 can inhibit p53-mediated ferroptosis, but not other tumor suppressors.

Thanks very much for the kind comments and great suggestions. As suggested, we had performed Co-IP assay with anti-HA antibodies to immunocapture NOLC1-FLAG. As shown in Fig. 5K, p53 DNA binding domain (DBD)-HA could immunocapture with NOLC1, further indicated that NOLC1 could binding to p53 DBD. Moreover, we concur with the reviewer that adding experiments using multiple p53 alterations, however considering that different p53 mutants have completely different functional changes. Therefore, we using siRNA to knockdown p53 level in MGC-803 cells, the results shown that NOLC1 mediated resistance was disappear and the GPX4 level was increased (Fig. S10). These data have shown that NOLC1 promotes GC resistance via mediated p53 functions.

(8) In Figure S5B, the LDH release can be blocked by Fer-1?

Thanks for raising this good question. As suggested, Fer-1 (20 μmol/mL) significantly blocked the LDH release in NOLC1 knockdown group (Fig S6E). This data further confirmed that NOLC1 suppressed Cis-induced ferroptosis.

(9) How about the ubiquitination assay in MGC-803 cells?

Thanks for raising this good question. As suggested, we also analyzed the ubiquitination assay in MGC-803 cells. As the result showed that NOLC1 also could increasing level of ubiquitination of p53 (Fig. 6H).

(10) In Figure 6H, the DBD domain of NOLC1 is required for inhibiting P53 ubiquitination.

Thanks for your opinion. However, in our paper, we only mentioned that p53 DBD domain, rather than NOLC1 DBD domain. Also, we did not find any DNA binding function of NOLC1 in the Pubmed database. Therefore, we would like to ask whether the revised opinion is correct.

(11) In Figure 8B, the CD3 antibody is not specific, please change it to a new one.

Thanks very much for the kind comments and great suggestions. As suggested, we have used new CD3 antibody and the new data was added in Fig. 8B.

(12) The authors report that NOLC1 influences peripheral blood lymphocytes with cisplatin treatment, with or without PD-1. Could the authors explain why NOLC1 would affect peripheral blood lymphocytes? Additionally, did they assess immune cell infiltration in the tumor microenvironment (TME) by flow cytometry?

Thanks for raising good question. The tumor size of the knockdown group treated with Cis + PD-1 was too small (less than 100 mg) to extract enough infiltrated immune cells (less than 10000 CD45^+^ cells), thus we chose to detect immune cells in the blood of the mice. Considering that the infiltrating immune cells including CTLs were originate from peripheral blood by circulation. Under the normal conditions, serval tumor biology behavior impact the TME to limit immune responses and present barriers to cancer therapy. For example, tumor could express or secret lots of negative regulator like PD-L1. Causing immune cells cannot recognize tumor cells and infiltrate into tumor tissue. Ferroptosis, as a new from of ICD, could damage tumor cell plasm and release amount of tumor associated antigen and tumor-specific antigens causing immune cells priming and activation. Eventually, the activated immune cells in peripheral blood travel towards the tumor site, infiltrating the tumor tissue under favorable co-stimulatory conditions and guided by chemokine gradients. Once within the tumor microenvironment, these activated T cells can control tumor growth through direct tumor cell destruction and cytokine-mediated processes [5–8]

To assess immune cell infiltration in the TME, we analyzed the tumor infiltrated CD3^+^ and CD8^+^ immune cells in tumor tissue by immunofluorescence (Fig. 8B). Thus, the peripheral blood lymphocytes could reflect the infiltration of immune cells in the tumor.

Minor concerns:

(1) Please clarify the statistical methods in each figure legend.

Thanks for your opinion. We have added statistical methods in each figure legend.

(2) In Figure 2D, please provide statistical data of cleaved-caspase3 expression.

Thanks for your opinion. As is shown in Fig. S5B-C, the relative cleaved-caspase3 were provided.

(3) Please ensure that the canonical expressions used in the research paper are adhered to.

Thanks for your opinion. We have carefully modified our expressions in our paper.

(4) Please pay more attention to the grammar and formatting of texts.

Thanks for your opinion. We revised our manuscript through the American Journal Experts (AJE) service.

**Reviewer #2:**
Summary:Shengsheng Zhao et al. investigated the role of nucleolar and coiled-body phosphoprotein 1 (NOLC1) in relegating gastric cancer (GC) development and cisplatin-induced drug resistance in GC. They found a significant correlation between high NOLC1 expression and the poor prognosis of GC. Meanwhile, upregulation of NOLC1 was associated with cis-resistant GC. Experimentally, the authors demonstrate that knocking down NOLC1 increased GC sensitivity to Cis possibly by regulating ferroptosis. Mechanistically, they found NOLC1 suppressed ferroptosis by blocking the translocation of p53 from the cytoplasm to the nucleus and promoting its degradation. In addition, The authors also evaluated the effect of combinational treatment of anti- PD-1 and cisplatin in NOLC1-knockdown tumor cells, revealing a potential role of NOLC1 in the targeted therapy for GC.Strengths:Chemoresistance is considered a major reason causing failure of tumor treatment and death of cancer patients. This paper explored the role of NOLC1 in the regulation of Cis-mediated resistance, which involves a regulated cell death named ferroptosis. These findings provide more evidence highlighting the study of regulated cell death to overcome drug resistance in cancer treatment, which could give us more potential strategies or targets for combating cancer.Weaknesses:More evidence supporting the regulation of ferroptosis induced by Cisplatin by NOLC1 should be added. Particularly, the role of ferroptosis in the cisplatin-resistance should be verified and whether NOLC1 regulates ferroptosis induced by additional FINs should be explored. Besides, the experiments to verify the regulation of ferroptosis sensitivity by NOLC1 are sort of superficial. The role of MDM2/p53 in ferroptosis or cisplatin resistance mediated by NOLC1 should be further studied by genetic manipulation of p53, which is the key evidence to confirm its contribution to NOLC1 regulation of GC and relative cell death.Major points:(1) More evidence supporting the regulation of ferroptosis induced by Cisplatin by NOLC1 should be added. Particularly, the role of ferroptosis in the cisplatin-resistance should be verified and whether NOLC1 regulates ferroptosis induced by additional FINs should be explored.

Thanks very much for the kind comments and great suggestions. As suggested, we have further analyzed the ferroptosis inhibit ability of NOLC1 in MGC-45 cells treated with Erastin, a common used ferroptosis activator. As shown in Fig. S6B, the ferroptosis activated by Erastin was also blocked by NOLC1.

(2) In Figure 1J, the CR cell line should obviously have less apoptosis-maker c-PARP expression, which means these cells are resistant to apoptosis induced by CR. Thus, it would be more rational to study the role of apoptosis regulation by NOLC1. Why did the later data shift to the study of ferroptosis?

Thanks for raising this good question. In the CR cells, the expression levels of many genes were changed, so it is uncertain whether the decreased expression level of cleaved-PARP in the resistant cells is caused by NOLC1 up-regulated. To explore the specific mechanism of NOLC1 mediated resistant, we performed the TEM imaging (Fig. 4A, S6A) and the results showed that cells exhibited classic ferroptosis morphological changes. Moreover, the BCL-2 (an anti-apoptotic protein, and regulated by p53 via protein interaction in cytoplasm) was increased after NOLC1 knockdown (Fig S5A). This phenomenon may cause by the increasing p53 levels in the cytoplasm[3,4] (Fig 5I). Taken together we shift to study of cisplatin induced ferroptosis.

(3) Besides, how about the regulation of apoptosis during cis-resistance by NOLC1 in GC?

Thanks for raising this good question. As mentioned above the Cis induced apoptosis was not as significant as ferroptosis, caused by BCL-2 (a key anti-apoptosis protein) increasing which is mediated by p53 via protein interaction in cytoplasm. NOLC1 increased plasm p53 level subsequently increased BCL-2 level.

(4) The experiments to verify the regulation of ferroptosis sensitivity by NOLC1 are sort of superficial. The role of MDM2/p53 in ferroptosis or cisplatin resistance mediated by NOLC1 should be further studied by genetic manipulation of p53, which is the key evidence to confirm its contribution to NOLC1 regulation of GC and relative cell death.

Thanks for raising this good question. As is shown in Fig S10, after knockdown p53 protein level by using siRNA, NOLC1 could not promote Cis-resistance and the GPX4 level was increased reflecting that NOLC1 promotes Cis resistance via mediate p53 function.

(5) In Figure 2, the data indicated that the knockdown of NOLC1 increased rH2Ax in the presence of Cisplatin, which indicated that NOLC1 might regulate DNA damage-related cellular function. These functions should be more relevant to cisplatin resistance, considering the fundamental effect of this chemo drug.

Thanks very much for the kind comments and great suggestions. Indeed, we found that DNA damage was more obvious in knockdown groups, but the ferroptotic changes like ROS and mitochondrial membrane damage were also significantly different in knockdown groups. Considering that as a chemo drug, cisplatin not only induces damage DNA but also acts as a stress which could activates various signal pathways including apoptosis, ferroptosis, pyroptosis, necroptosis, etc., under different drug concentrate or time [9–11]. Therefore, it is important to find out the NOLC1 predominantly blocked pathway in GC.

(6) In Figure.4, ferroptosis inhibitors like Ferr-1 or DFO should be used to verify the regulation of ferroptosis by Cisplatin and NOLC1.

Thanks very much for the kind comments and great suggestions. As suggested, we performed additional LDH release assay. The results showed that Fer-1 also could block cisplatin induced LDH release in NOLC1 knockdown groups (Fig. S6E).

(7) In Figure 4H, Cisplatin decreased FSP1 and GPX4, which could be enhanced in the NOLC1-konckdown cell line. Meanwhile, the knockdown of NOLC1 increased the ACSL4 level. These findings could be the key reason for the regulation of ferroptosis by NOLC1 rather than p53 since they all are direct regulators of ferroptosis.

Thanks very much for the kind comments and great suggestions. We rewrote the text as you suggested. Recently, it also has been reported that ACSL4-regulated ferroptosis is related to p53, but the exact mechanism is still unclear [12]. Moreover, further studies of specific relation between NOLC1 and FSP1/ACSL4 will be conducted in the further

(8) Whether p53 mediates the regulation of ferroptosis and cisplatin resistance by NOLC1 should be thoroughly studied using p53-KO cell lines.

Thanks very much for the kind comments and great suggestions. As previously mentioned, by using si-RNA to knockdown p53, the NOLC1 mediate Cis-resistance were blocked (Fig. S10). Meanwhile, the GPX4 level was also increased in p53/NOLC1 double-knockdown groups compared to the NOLC1 knockdown group. These data indicating that NOLC1 suppresses ferroptosis via mediating p53 functions.

**Reviewer #3:**
The authors have put forth a compelling argument that NOLC1 is indispensable for gastric cancer resistance in both in vivo and in vitro models. They have further elucidated that NOLC1 silencing augments cisplatin-induced ferroptosis in gastric cancer cells. The mechanistic underpinning of their findings suggests that NOLC1 modulates the p53 nuclear/plasma ratio by engaging with the p53 DNA Binding Domain, which in turn impedes p53-mediated transcriptional regulation of ferroptosis. Additionally, the authors have shown that NOLC1 knockdown triggers the release of ferroptosis-induced damage-associated molecular patterns (DAMPs), which activate the tumor microenvironment (TME) and enhance the efficacy of the anti-PD-1 and cisplatin combination therapy.Strengths:The manuscript presents a robust dataset that substantiates the authors' conclusion. They have identified NOLC1 as a potential oncogene that confers resistance to immuno-chemotherapy in gastric cancer through the mediation of ferroptosis and subsequent TME reprogramming. This discovery positions NOLC1 as a promising therapeutic target for gastric cancer treatment. The authors have delineated a novel mechanistic pathway whereby NOLC1 suppresses p53 transcriptional functions by reducing its nuclear/plasma ratio, underscoring the significance of p53 nuclear levels in tumor suppression over total protein levels.Weaknesses:While the overall findings are commendable, there are specific areas that could benefit from further refinement. The authors have posited that NOLC1 suppresses p53- mediated ferroptosis; however, the mRNA levels of ferroptosis genes regulated by p53 have not been quantified, which is a critical gap in the current study. In Figure 4A, transmission electron microscopy (TEM) results are reported solely for the MGC-803 cell line. It would be beneficial to include TEM data for the MKN-45 cell line to strengthen the findings. The authors have proposed a link between NOLC1-mediated reduction in the p53 nuclear/plasma ratio and gastric cancer resistance, yet the correlation between this ratio and patient prognosis remains unexplored, which is a significant limitation in the context of clinical relevance.

Thanks very much for the kind comments and great suggestions. As suggested, recently studies have reported that CDKN1A (also called p21, a p53 transcriptional mediated protein) could promotes ferroptosis[13], the mRNA levels of ferroptosis genes regulated by p53 have were quantified in Fig. S8G-H. Moreover, we further proceed TEM imaging in MKN-45 cells, the result was consistent to MGC-803 cells, reflecting that NOLC1 has a broad spectrum of promoting drug resistance in gastric cancer. Also, recently studies have reported that p53 transcriptional active and p53 transcriptional inactive types include patients with intermediate prognosis and recurrence rates, with the p53-acvtie group showing better prognosis[14]. Considering p53 transcriptional activity depends on p53 nuclear accumulation, we assume that the low level of p53 nuclear/plasma may cause poor prognosis in gastric cancer. Meanwhile we will further collect enough samples and their prognostic information to analysis NOLC1-mediated reduction in the p53 nuclear/plasma ratio and gastric cancer resistance.

References

(1) Z. Seferbekova, A. Lomakin, L.R. Yates, M. Gerstung, Spatial biology of cancer evolution, Nat Rev Genet 24 (2023) 295–313. https://doi.org/10.1038/s41576-022-00553-x.

(2) T. Matsuoka, M. Yashiro, Molecular Mechanism for Malignant Progression of Gastric Cancer Within the Tumor Microenvironment, IJMS 25 (2024) 11735. https://doi.org/10.3390/ijms252111735.

(3) Y. Liu, Z. Su, O. Tavana, W. Gu, Understanding the complexity of p53 in a new era of tumor suppression, Cancer Cell (2024) S1535610824001338. https://doi.org/10.1016/j.ccell.2024.04.009.

(4) R. Pan, V. Ruvolo, H. Mu, J.D. Leverson, G. Nichols, J.C. Reed, M. Konopleva, M. Andreeff, Synthetic Lethality of Combined Bcl-2 Inhibition and p53 Activation in AML: Mechanisms and Superior Antileukemic Efficacy, Cancer Cell 32 (2017) 748-760.e6. https://doi.org/10.1016/j.ccell.2017.11.003.

(5) E. Catanzaro, M. Beltrán-Visiedo, L. Galluzzi, D.V. Krysko, Immunogenicity of cell death and cancer immunotherapy with immune checkpoint inhibitors, Cell Mol Immunol 22 (2024) 24–39. https://doi.org/10.1038/s41423-024-01245-8.

(6) G. Lei, L. Zhuang, B. Gan, The roles of ferroptosis in cancer: Tumor suppression, tumor microenvironment, and therapeutic interventions, Cancer Cell 42 (2024) 513–534. https://doi.org/10.1016/j.ccell.2024.03.011.

(7) E. Catanzaro, R. Demuynck, F. Naessens, L. Galluzzi, D.V. Krysko, Immunogenicity of ferroptosis in cancer: a matter of context?, Trends in Cancer 10 (2024) 407–416. https://doi.org/10.1016/j.trecan.2024.01.013.

(8) X. Jiang, B.R. Stockwell, M. Conrad, Ferroptosis: mechanisms, biology and role in disease, Nat Rev Mol Cell Biol 22 (2021) 266–282. https://doi.org/10.1038/s41580-020-00324-8.

(9) J.-L. Roh, E.H. Kim, H. Jang, D. Shin, Nrf2 inhibition reverses the resistance of cisplatin-resistant head and neck cancer cells to artesunate-induced ferroptosis, Redox Biology 11 (2017) 254–262. https://doi.org/10.1016/j.redox.2016.12.010.

(10) X. Wang, Y. Zhou, D. Wang, Y. Wang, Z. Zhou, X. Ma, X. Liu, Y. Dong, Cisplatin-induced ototoxicity: From signaling network to therapeutic targets, Biomedicine & Pharmacotherapy 157 (2023) 114045. https://doi.org/10.1016/j.biopha.2022.114045.

(11) J. Liang, G. Bi, Y. Huang, G. Zhao, Q. Sui, H. Zhang, Y. Bian, J. Yin, Q. Wang, Z. Chen, C. Zhan, MAFF confers vulnerability to cisplatin-based and ionizing radiation treatments by modulating ferroptosis and cell cycle progression in lung adenocarcinoma, Drug Resistance Updates 73 (2024) 101057. https://doi.org/10.1016/j.drup.2024.101057.

(12) M.Y. Kosim, T. Fukazawa, M. Miyauchi, N. Hirohashi, K. Tanimoto, p53 status modifies cytotoxic activity of lactoferrin under hypoxic conditions, Front. Pharmacol. 13 (2022) 988335. https://doi.org/10.3389/fphar.2022.988335.

(13) Q. Gao, J. Chen, C. Li, J. Zhan, X. Yin, B. Li, H. Dong, L. Luo, Z. Li, CDKN1A promotes Cis-induced AKI by inducing cytoplasmic ROS production and ferroptosis, Food and Chemical Toxicology 193 (2024) 115003. https://doi.org/10.1016/j.fct.2024.115003.

(14) R. Cristescu, Molecular analysis of gastric cancer identifies subtypes associated with distinct clinical outcomes, Nature Medicine (2015).